# Tom70-based transcriptional regulation of mitochondrial biogenesis and aging

Qingqing Liu[1], Catherine E Chang[1], Alexandra C Wooldredge[1], Benjamin Fong[1], Brian K Kennedy[1,2,3,4], Chuankai Zhou[1,5]*

[1]Buck Institute for Research on Aging, Novato, United States; [2]Healthy Longevity Programme, Yong Loo Lin School of Medicine, National University of Singapore, Singapore, Singapore; [3]Centre for Healthy Longevity, National University Health System, Singapore, Singapore; [4]Singapore Institute of Clinical Sciences, A(*)STAR, Singapore, Singapore; [5]USC Leonard Davis School of Gerontology, University of Southern California, Los Angeles, United States

**Abstract** Mitochondrial biogenesis has two major steps: the transcriptional activation of nuclear genome-encoded mitochondrial proteins and the import of nascent mitochondrial proteins that are synthesized in the cytosol. These nascent mitochondrial proteins are aggregation-prone and can cause cytosolic proteostasis stress. The transcription factor-dependent transcriptional regulations and the TOM-TIM complex-dependent import of nascent mitochondrial proteins have been extensively studied. Yet, little is known regarding how these two steps of mitochondrial biogenesis coordinate with each other to avoid the cytosolic accumulation of these aggregation-prone nascent mitochondrial proteins. Here, we show that in budding yeast, Tom70, a conserved receptor of the TOM complex, moonlights to regulate the transcriptional activity of mitochondrial proteins. Tom70's transcription regulatory role is conserved in *Drosophila*. The dual roles of Tom70 in both transcription/biogenesis and import of mitochondrial proteins allow the cells to accomplish mitochondrial biogenesis without compromising cytosolic proteostasis. The age-related reduction of Tom70, caused by reduced biogenesis and increased degradation of Tom70, is associated with the loss of mitochondrial membrane potential, mtDNA, and mitochondrial proteins. While loss of Tom70 accelerates aging and age-related mitochondrial defects, overexpressing TOM70 delays these mitochondrial dysfunctions and extends the replicative lifespan. Our results reveal unexpected roles of Tom70 in mitochondrial biogenesis and aging.

*For correspondence:
kzhou@buckinstitute.org

Competing interest: The authors declare that no competing interests exist.

## Editor's evaluation

The authors test the hypothesis that components of the TOM complex regulate efficient mitochondrial biogenesis by coordinating the synthesis (via controlling transcription of the corresponding RNAs) of mitochondrial proteins with the rate of mitochondrial protein import. It has previously been established that failure to import mitochondrial proteins results in the accumulation of toxic protein aggregates in the cytosol. The authors conclude that Tom70 fulfills this role and find that Tom70 expression declines as cells age, which contributes to age-associated mitochondrial dysfunction.

## Introduction

Mitochondria play critical roles in a growing list of cellular activities, such as signaling, metabolism, and inflammation. A decline in mitochondrial function, both in quality and quantity, has been associated with normal aging and many age-related diseases (*Gureev et al., 2019*; *Sun et al., 2016*). Mitochondrial quality and quantity are determined by mitochondrial biogenesis and quality control mechanisms

(*Baker et al., 2011*; *Pickles et al., 2018*; *Shpilka and Haynes, 2018*). More than 99% of mitochondrial proteins are encoded by the nuclear genome (*Fox, 2012*; *Mootha et al., 2003*; *Morgenstern et al., 2017*; *Pagliarini et al., 2008*; *Vögtle et al., 2017*). The expression of mitochondrial proteins is mainly regulated at the transcriptional level by transcription factors that integrate cellular and environmental signals (*de Winde and Grivell, 1993*; *Jornayvaz and Shulman, 2010*; *Scarpulla, 2008*). For example, in budding yeast, cells sense nutrients and regulate the expression of mitochondrial proteins through Hap4, a well-studied transcription activator of mitochondrial biogenesis (*Forsburg and Guarente, 1989*; *McNabb et al., 1995*; *Olesen et al., 1987*; *Pinkham and Guarente, 1985*; *Santangelo, 2006*; *Schüller, 2003*). In addition to transcription activation, repressors such as Mig1 and Rox1 also play a critical role to set the balance and maintain optimal expression of mitochondrial proteins in response to different growth conditions (*Lowry and Zitomer, 1984*; *Santangelo, 2006*; *Schüller, 2003*; *Ter Linde and Steensma, 2002*; *Trueblood et al., 1988*).

After completing transcription, these mRNAs encoding mitochondrial proteins are translated in the cytosol and imported by the TOM-TIM complex (*Wiedemann and Pfanner, 2017*). Most mitochondrial proteins are selectively recognized and imported via Tom20 and Tom70, two major receptors of the TOM complex. The Tom20 pathway is known to import nascent mitochondrial proteins that contain an N-terminal signal peptide. A significant number of nascent mitochondrial proteins lack the typical N-terminal peptides and, instead, present the internal mitochondrial targeting signals (iMTS) that span the entire primary sequence and are imported via the Tom70 pathway (*Backes et al., 2018*; *Sirrenberg et al., 1996*). Another unique feature of the Tom70 pathway is that cytosolic chaperones, such as Hsp70, are involved in maintaining these mitochondrial proteins in unfolded states before being transferred to Tom70 for import (*Hoseini et al., 2016*; *Young, 2003*; *Zanphorlin et al., 2016*). Therefore, mitochondrial import, especially the Tom70-mediated pathway, is integrated into the cytosolic proteostasis. Indeed, the accumulation of nascent mitochondrial proteins in the cytosol triggers cytosolic stress responses, known as UPRam and mPOS (*Wang and Chen, 2015*; *Wrobel et al., 2015*). Given the aggregation-prone nature of these nascent mitochondrial proteins (*Nowicka et al., 2021*), a balance between the production and import of mitochondrial proteins is required to avoid the accumulation of these proteins in the cytosol.

Although the transcriptional regulations of mitochondrial proteins and the molecular mechanisms of mitochondrial import have been extensively studied in the past, they were investigated separately, and little is known regarding how these two steps of mitochondrial biogenesis coordinate with each other to avoid the cytosolic accumulation of these aggregation-prone nascent mitochondrial proteins. In this study, we use budding yeast as the model to provide evidence that the mitochondrial import receptor Tom70 moonlights to regulate the transcriptional activity of mitochondrial proteins. This transcription regulatory role of Tom70 is conserved in *Drosophila melanogaster* (fruit fly). Our results suggest that cells use the same molecule, Tom70, to regulate both the transcription/biogenesis and import of mitochondrial proteins so the nascent mitochondrial proteins do not compromise cytosolic proteostasis or cause cytosolic protein aggregation. Importantly, the Tom70 abundance is decreased during aging, which is associated with age-dependent loss of mitochondrial membrane potential, mtDNA, and mitochondrial proteins. While loss of Tom70 accelerates aging and age-related mitochondrial defects, overexpressing TOM70 delays these mitochondrial dysfunctions and extends the replicative lifespan. Therefore, the reduction of Tom70 is a key event contributing to the mitochondrial biogenesis and defects during aging.

## Results

### Tom70 regulates the transcriptional activity of mitochondrial proteins

As the abundance of TOM complex on the mitochondrial outer membrane determines how fast nascent mitochondrial proteins can be imported, we started out by examining if TOM proteins may also regulate the biogenesis of mitochondrial proteins so the biogenesis and import of nascent mitochondrial proteins can be coordinated with each other. We used the pGAL promoter to overexpress (OE) different TOM proteins and Hap4, which is known to drive mitochondrial biogenesis (*Lin et al., 2002*). To examine their effects on mitochondrial biogenesis, we randomly selected four mitochondrial proteins to represent different sub-compartments of the organelle. The expression levels of representative mitochondrial proteins were visualized by endogenous C-terminal GFP tagging and

expressed from their native promoters using the strains from the yeast GFP library (*Huh et al., 2003*). This library has been extensively characterized against different proteomic results to show high accuracy and fidelity (*Ho et al., 2018*; *Newman et al., 2006*). To quantitatively compare the GFP signals from different strains, we imaged the cells with an avalanche photodiode (APD), which is a single-photon sensitive detector (*Bruschini et al., 2019*). As a single GFP protein emits ~21 photons in our imaging setting (*Kubitscheck et al., 2000*), this single-photon sensitive detector allowed us to accurately compare the abundance of these mitochondrial proteins across different conditions in live cells.

Compared to the wild-type cells cultured in the same medium, we found that HAP4 OE indeed could increase the expression of two out of these four randomly chosen mitochondrial proteins (*Figure 1A*, *Figure 1—figure supplement 1A*). This is consistent with previous results that HAP4 OE increases the biogenesis of roughly 1/4 of the mitochondrial proteome (255 out of >1000 mitochondrial proteins) (*Lin et al., 2002*). Among these TOM proteins, only TOM70 OE increased the abundance of these mitochondrial proteins (*Figure 1A*; *Figure 1—figure supplement 1A*). Compared to HAP4 OE, TOM70 OE has a similar but broader effect on these mitochondrial proteins (*Figure 1A*; *Figure 1—figure supplement 1A*). Interestingly, Tom71, the paralog of Tom70, failed to induce the expression of any of these mitochondrial proteins (*Figure 1A*; *Figure 1—figure supplement 1A*). The effect of TOM70 OE on these mitochondrial proteins is not caused by or unique to galactose: (1) all strains, including wild type and HAP4 OE, were cultured in the same galactose medium while OE of Tom70, but not other TOM proteins, increased mitochondrial biogenesis; (2) similar results were observed when Tom70 was overexpressed from an artificial promoter Z3EV, which is inducible by the β-estradiol, in the glucose medium (*Figure 1—figure supplement 1B*; *McIsaac et al., 2013*). We then expanded the list to cover more mitochondrial proteins from all four sub-compartments of the organelle and found TOM70 OE generally increased their abundance (*Figure 1B*).

The enhanced mitochondrial biogenesis in response to TOM70 OE was also observed at the transcriptional level: TOM70 OE generally increased the mRNA levels of nucleus-encoded mitochondrial proteins, including the tail-anchored small TOM proteins that are known to be post-translationally inserted into the mitochondrial outer membrane independently of Tom20 and Tom70 (*Figure 1C*; *Drwesh and Rapaport, 2020*). This indicates that the transcriptional activation of mitochondrial biogenesis upon TOM70 OE is not restricted to Tom70's substrates. Consistent with its role in transcription activation, when Tom70 was selectively removed from mitochondria by TEV protease, the mRNA level of many mitochondrial proteins reduced (*Figure 1C*, *Figure 1—figure supplement 1C*). This transcription regulatory role of Tom70 is conserved as an enhanced mitochondrial biogenesis was observed in fruit fly larvae when UAS-TOM70 was overexpressed in muscles by Mef2-Gal4 (*Figure 1C*, *Figure 1—figure supplement 1D*). As overexpressing Tom70's cytosolic domain alone, which stay diffusively in the cytosol, did not activate the transcription of these mitochondrial proteins (*Figure 1C*), these results suggest that Tom70 has to be on the mitochondrial outer membrane to control the transcriptional activity of many mitochondrial proteins.

Which transcription factor(s) act downstream of Tom70? Retrograde signaling from mitochondria, marked by the increased nuclear-to-cytosol ratio of transcription factors Rtg1/3, is a known stress response toward mitochondrial stress and defects (*Liu and Butow, 2006*). However, the nuclear/cytosol ratio of Rtg1/3-GFP in TOM70 OE cells is similar to the wild-type cells cultured in the same galactose medium (*Figure 1D*, *Figure 1—figure supplement 1E*). Additionally, knocking out Rtg3 failed to prevent the activation of mitochondrial biogenesis program in TOM70 OE cells (*Figure 1—figure supplement 1 F*). These results suggest that Tom70 regulates the transcription of mitochondrial proteins through transcription factors (TFs) other than the well-known retrograde signaling pathway. To search for the TFs downstream of Tom70, *in silico* analysis was conducted to find common TFs that recognize/bind the promoters of these mitochondrial genes (*Monteiro et al., 2020*). Some of these predicted TFs have expression changes upon TOM70 OE, indicating that Tom70 regulates these TFs (*Figure 2A*). A few of these TFs are so far not known to regulate mitochondrial biogenesis or metabolism (*Figure 2B*; *Bähler, 2005*). For example, Fkh1/2 are members of the Forkhead family transcription factor that are previously known to regulate the expression of G2/M phase genes (*Bähler, 2005*; *Martins et al., 2016*; *Postnikoff et al., 2012*). Among these TFs that responded to Tom70 OE, overexpressing some TFs, including Fkh1/2, can mimic the TOM70 OE effect and increase the abundance of mitochondrial proteins from different sub-compartments, such as Adh3, Qcr7, and Tim44 (*Figure 2C and D*). We asked whether Fkh1/2 are required for the Tom70-related

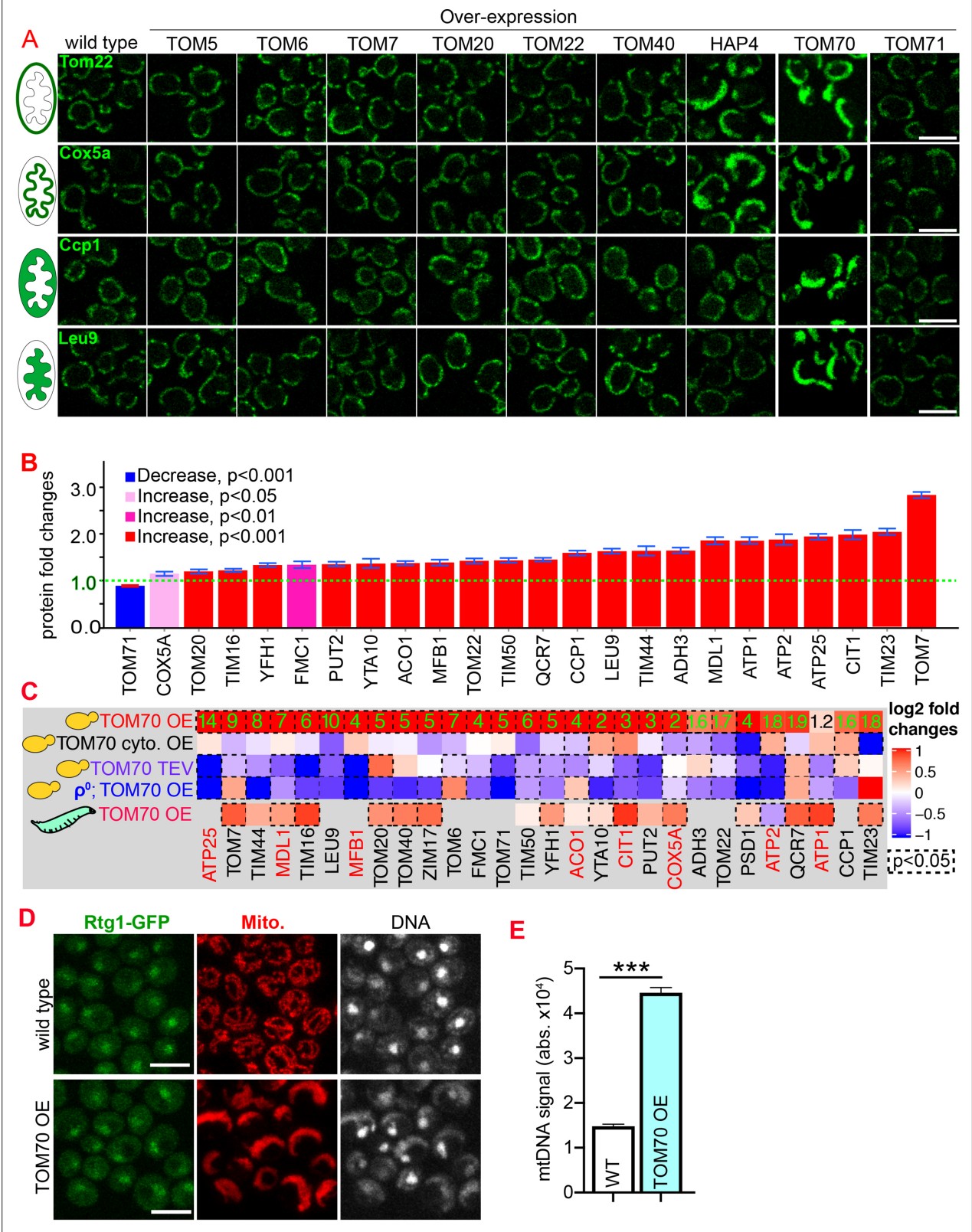

**Figure 1.** Tom70 regulates the transcriptional activity of mitochondrial proteins. (**A**) Representative images for proteins localized to different sub-compartments of mitochondria in cells overexpressing (OE) different TOM proteins and HAP4 from pGal promoter. HAP4 OE increased the biogenesis of some, but not all, mitochondrial proteins. Mitochondrial proteins were visualized by endogenous C-terminal GFP tagging and expressed from their native promoters. As both wild type and OE strains were cultured in the same medium, the mitochondrial biogenesis effect of TOM70 OE and HAP4

*Figure 1 continued on next page*

*Figure 1 continued*

OE is not due to the presence of galactose. (**B**) Quantification of mitochondrial protein levels in TOM70 OE cells normalized to wild-type cells cultured in the same medium. Mitochondrial proteins were visualized and quantified by endogenous C-terminal GFP tagging. These mitochondrial proteins were randomly chosen from the yeast GFP library based on their localization in mitochondria to represent all four sub-compartments of mitochondria. Bar graphs are the normalized mean and s.e.m. Data were analyzed with unpaired two-tailed t test. (**C**) RT-qPCR quantification of the mRNA abundance for different mitochondrial proteins in yeast and fruit fly strains with different levels of mitochondrial Tom70. TOM70 cyto. OE, overexpression of Tom70 cytosolic domain without transmembrane domain; TOM70 OE, TOM70 overexpression; TOM70 TEV, the cytosolic domain of Tom70 was removed by TEV protease. $\rho^0$, petite cells without mtDNA. For yeast, cells with the same genetic background but lacking TOM70 TEV or TOM70 OE were used as control. For example, mRNA from $\rho^0$/TOM70 OE cells were normalized to $\rho^0$ cells lacking TOM70 OE. All different yeast strains, including the controls, were cultured in the same galactose medium. For fruit fly, 3rd instar larvae from control (UAS-TOM70) and TOM70 OE (Mef2-Gal4; UAS-TOM70) were used to extract mRNA from the whole animal. The mRNA abundance from each strain was normalized to control and the average fold changes from three replicates are shown. Only conserved orthologous genes from fly are included (https://www.alliancegenome.org/). Validated substrates of Tom70 are colored in red text (***Backes et al., 2018***; ***Kondo-Okamoto and Shaw, 2008***; ***Melin et al., 2015***; ***Yamamoto et al., 2009***; ***Young, 2003***). Dash line boxes indicate the ones with p < 0.05 from unpaired two-tailed t test. The inserted green numbers are fold changes of each protein in TOM70 OE cells. (**D**) Representative images of Rtg1-GFP in control and TOM70 OE cells. Rtg1-GFP, endogenous C-terminal GFP tagging of Rtg1. DNA was stained with Hoechst dye. All different yeast strains were cultured in the same galactose medium. (**E**) Quantification of mtDNA abundance from Hoechst staining in (**D**). 1116 and 1090 cells quantified for each. Data were analyzed with unpaired two-tailed t test: ***, p < 0.001. Scale bar for all images: 5 μm. Images are representative of at least two independent experiments. Sample sizes in B are given in ***Supplementary file 4***.

The online version of this article includes the following figure supplement(s) for figure 1:

**Figure supplement 1.** Additional data demonstrating that Tom70 regulates the transcription of mitochondrial proteins.

mitochondrial biogenesis program. When FKH1 or FKH2 are deleted, the induced expression of mitochondrial proteins in TOM70 OE cells is partially blocked (***Figure 2E***). We noticed that none of the Fkh1/2 mutants completely blocked the effect of TOM70 OE, indicating that Tom70 can signal through multiple TFs to regulate mitochondrial biogenesis. This is consistent with the observation that TOM70 OE has broader effects on mitochondrial biogenesis than OE of a single TF, such as HAP4 OE (***Figure 1A***).

We also tested whether reactive oxygen species (ROS) could mediate the downstream signaling of TOM70 OE as low levels of ROS were proposed to have important roles in mitochondrial retrograde signaling and mitochondrial hormesis (***Choi et al., 2017***; ***Ristow and Schmeisser, 2014***; ). Similar to the knockout of Fkh1/2, treating cells with ROS scavenger N-acetylcysteine (NAC) partially affected the mitochondrial biogenesis induction in TOM70 OE cells (***Figure 2—figure supplement 1***). These data are consistent with the model that Tom70 regulates the mitochondrial biogenesis via multiple pathways and TFs. Disrupting one factor at a time only affects a specific pathway and TF, therefore, has limited effect on some, but not all, mitochondrial proteins.

## Tom70 regulates the abundance of mtDNA

Using single-photon-sensitive APD and quantitative DNA staining (***Gomes et al., 2018***), we found that in addition to the mitochondrial proteins, TOM70 OE also increased mtDNA significantly (***Figure 1D and E***). In contrast to TOM70 OE, knocking out TOM70 led to the reduction of mtDNA (***Figure 3A and B***). Therefore, Tom70 regulates the biogenesis of both mitochondrial proteins and mtDNA. DNA polymerase γ (Mip1 in yeast) is responsible for the replication and affects the abundance of mtDNA (***Foury, 1989***). We noticed that Mip1 contains the iMTS which is known to use Tom70 for mitochondrial import (***Figure 3C***; ***Backes et al., 2018***). Indeed, the abundance of Mip1 protein was increased in TOM70 OE cells (***Figure 3D***), which likely contributes to the observed increase of mtDNA. Consistently, MIP1 OE increased the abundance of mtDNA (***Figure 3E***), albeit to a lesser extent than TOM70 OE (***Figure 1E***), suggesting that additional factors also contribute to the increase of mtDNA in TOM70 OE cells. Interestingly, when mtDNA was removed, TOM70 OE could not activate the transcription of these nDNA-encoded mitochondrial proteins ($\rho^0$;TOM70 OE in ***Figure 1C***). This suggests that mtDNA is required for the Tom70-mediated transcriptional activation of nDNA-encoded mitochondrial proteins. Although TOM70 OE also increased the expression of basic mitochondrial transcription machineries, such as Rpo41 and Mtf1, upregulating their expressions could not increase the biogenesis of other mitochondrial proteins as TOM70 OE (***Figure 3—figure supplement 1A,B,C***). In contrast, we noticed that the mitochondrial translation is partially required for the biogenesis of some mitochondrial proteins in TOM70 OE cells (***Figure 3—figure supplement 1D***). Therefore, the role of

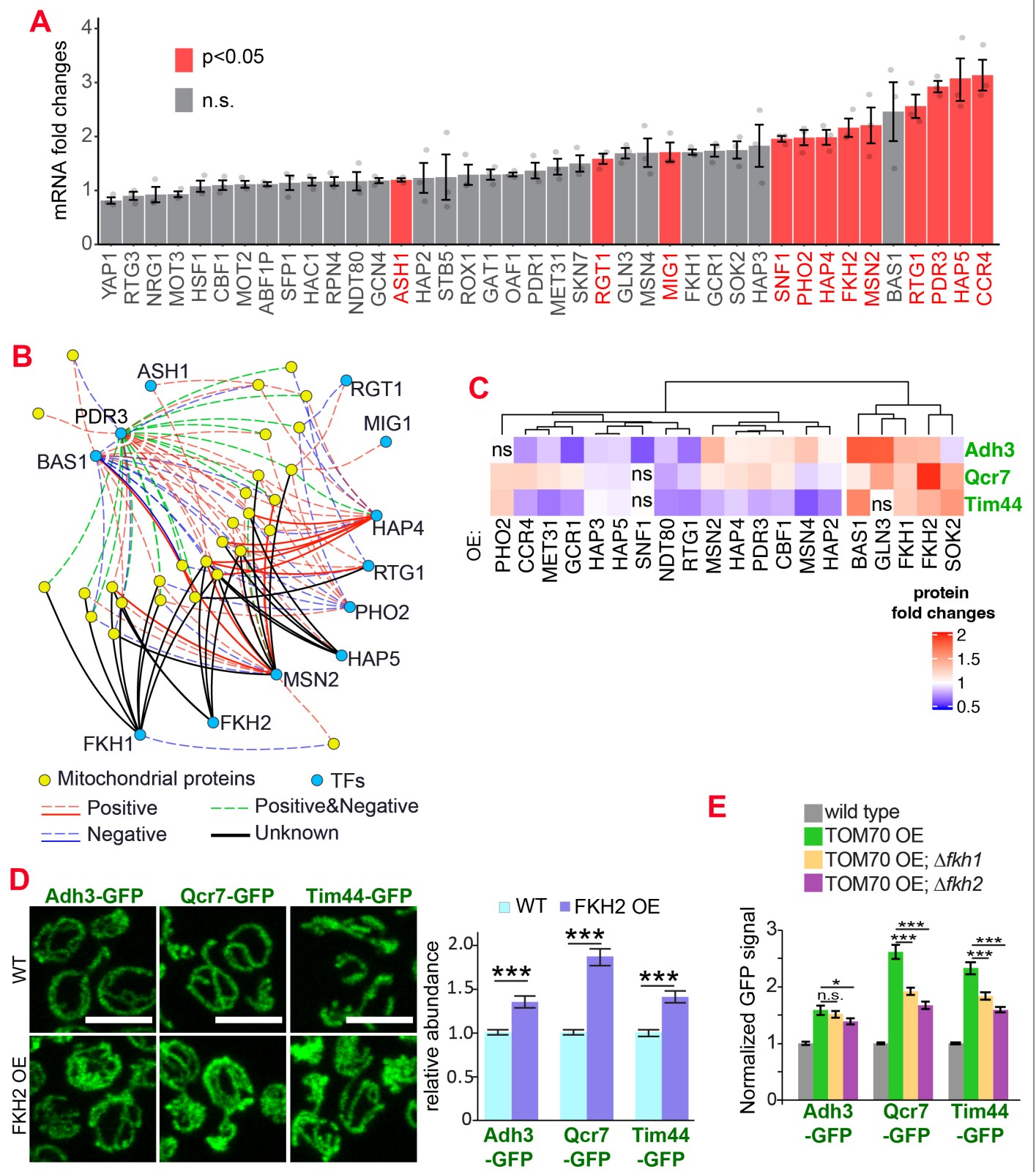

**Figure 2.** The transcription factors involved in the Tom70-dependent transcriptional regulation of mitochondrial proteins. (**A**) mRNA abundance of different transcription factors (TF) quantified by RT-qPCR in TOM70 OE strain and normalized to control cells. All different yeast strains, including wild-type control, were cultured in the same medium. Shown are the mean and s.e.m. from three replicates. (**B**) TF regulatory network predicted for these mitochondrial proteins tested in **Figure 1B** (**Monteiro et al., 2020**). Only the TFs that increase significantly in (**A**) is included to simplify the illustration.

*Figure 2 continued on next page*

*Figure 2 continued*

The positive and negative regulations from existing literature are shown with red and blue lines, respectively; green lines represent the ones have both positive and negative effects. Solid lines represent the DNA binding evidence and the dash lines for co-expression evidence. The predicted regulations based on the consensus sequences but have no previous experimental data are shown as black lines. (**C**) Expression changes of mitochondrial proteins from different sub-compartments upon overexpression of different TFs. The GFP signal of Adh3-GFP, Qcr7-GFP, and Tim44-GFP were quantified from TF OE cells and normalized to wild type control of each protein. p < 0.05 for all of them except for four that labeled as 'ns'. Data were analyzed with unpaired two-tailed t test. Both control and TOM70 OE strains were cultured in SC-raffinose medium and added 2% galactose for 5 hr before sample collection. (**D**) Representative images and quantification of different mitochondrial proteins in wild type and FKH2 OE strain. All different yeast strains were cultured in the same medium. (**E**) FKH1 and FKH2 are partially required for the mitochondria biogenesis program downstream of TOM70 OE. GFP signal of mitochondrial proteins were quantified in different strains and normalized to wild type. Scale bar for all images: 5 µm. Mitochondrial proteins were visualized by endogenous C-terminal GFP tagging and expressed from their own promoters. Images are representative of at least two independent experiments. Bar graphs are the normalized mean and s.e.m. Data were analyzed with unpaired two-tailed t test: ***, p < 0.001; *, p < 0.05; n.s., not significant. Sample sizes in (**D, E**) are given in *Supplementary file 4*.

The online version of this article includes the following figure supplement(s) for figure 2:

**Figure supplement 1.** ROS is partially required for the mitochondrial biogenesis program downstream of TOM70 OE.

mtDNA in the Tom70-regulated mitochondrial biogenesis can be partially attributed to the requirement of some mtDNA-encoded proteins.

## The Tom70-dependent regulation of mitochondrial biogenesis is involved in the cellular response to the mitochondrial import defect

Our results indicate that Tom70 not only controls the import of mitochondrial proteins, but also regulates their biogenesis. The nascent mitochondrial proteins synthesized in the cytosol are aggregation-prone and can cause cytosolic proteostasis stress and protein aggregation if not imported in a timely manner (*Nowicka et al., 2021*; *Wang and Chen, 2015*; *Wrobel et al., 2015*; *Figure 4—figure supplement 1A*). It has been shown that impaired mitochondrial import triggers cellular responses to reduce the biogenesis of mitochondrial proteins in order to achieve a new balance between biogenesis and import of nascent mitochondrial proteins, alleviating their cytosolic accumulation and aggregation (*Samluk et al., 2018*; *Topf et al., 2019*; *Wrobel et al., 2015*). Consistent with these previous reports, no protein aggregation was detected when we blocked mitochondrial import by shifting the *tim23^ts* cells to the restricted temperature, which impaired the TIM complex (*Figure 4A and C*; *Pareek et al., 2013*).

Given Tom70's dual roles in mitochondrial import and transcriptional regulation, we considered the possibility that Tom70 is involved in the cellular response to the impairment of mitochondrial import (*Figure 4A and B*). To test if Tom70 is required for the repression of mitochondrial biogenesis in *tim23^ts* cells, we knocked out Tom70 and found that *tim23^ts* cells lacking Tom70 formed large scale cytosolic protein aggregates upon Tim23 inactivation (*Figure 4C*). This suggests that Tim23-inactivation signals through Tom70 to repress the biogenesis of mitochondrial proteins and re-balance the biogenesis and import. As TOM70 OE upregulates transcriptional activity of mitochondrial proteins, we tested whether overexpressing Tom70 can overwrite the repressive cellular program of mitochondrial biogenesis induced by *tim23^ts* and cause cytosolic aggregation. Indeed, when we overexpressed Tom70 upon *tim23^ts* inactivation, cells failed to maintain cytosolic homeostasis and showed large-scale cytosolic protein aggregation (*Figure 4C*). In contrast, TOM70 OE alone did not have this effect, suggesting that the increase of mitochondrial biogenesis in TOM70 OE was balanced by enhanced mitochondrial import capacity (*Figure 4C*). This is consistent with the increase of many TOM and TIM proteins in TOM70 OE cells (*Figure 1C*). Next, we asked if this overwrite of repressive cellular program and loss of balance between biogenesis-import is a general outcome of any pathways/factors that induce mitochondrial biogenesis. However, HAP4 OE, which also controls mitochondrial biogenesis, did not cause cytosolic protein aggregation upon *tim23^ts*, suggesting that the Tom70-dependent mitochondrial biogenesis pathway plays a special role in the repressive cellular program induced by *tim23^ts*. Together, these results are consistent with a model that Tom70 is involved in the cellular response to the impairment of mitochondrial import in *tim23^ts* cells and regulate the biogenesis of mitochondrial proteins (*Figure 4—figure supplement 1B*; *Samluk et al., 2018*; *Topf et al., 2019*; *Wrobel et al., 2015*).

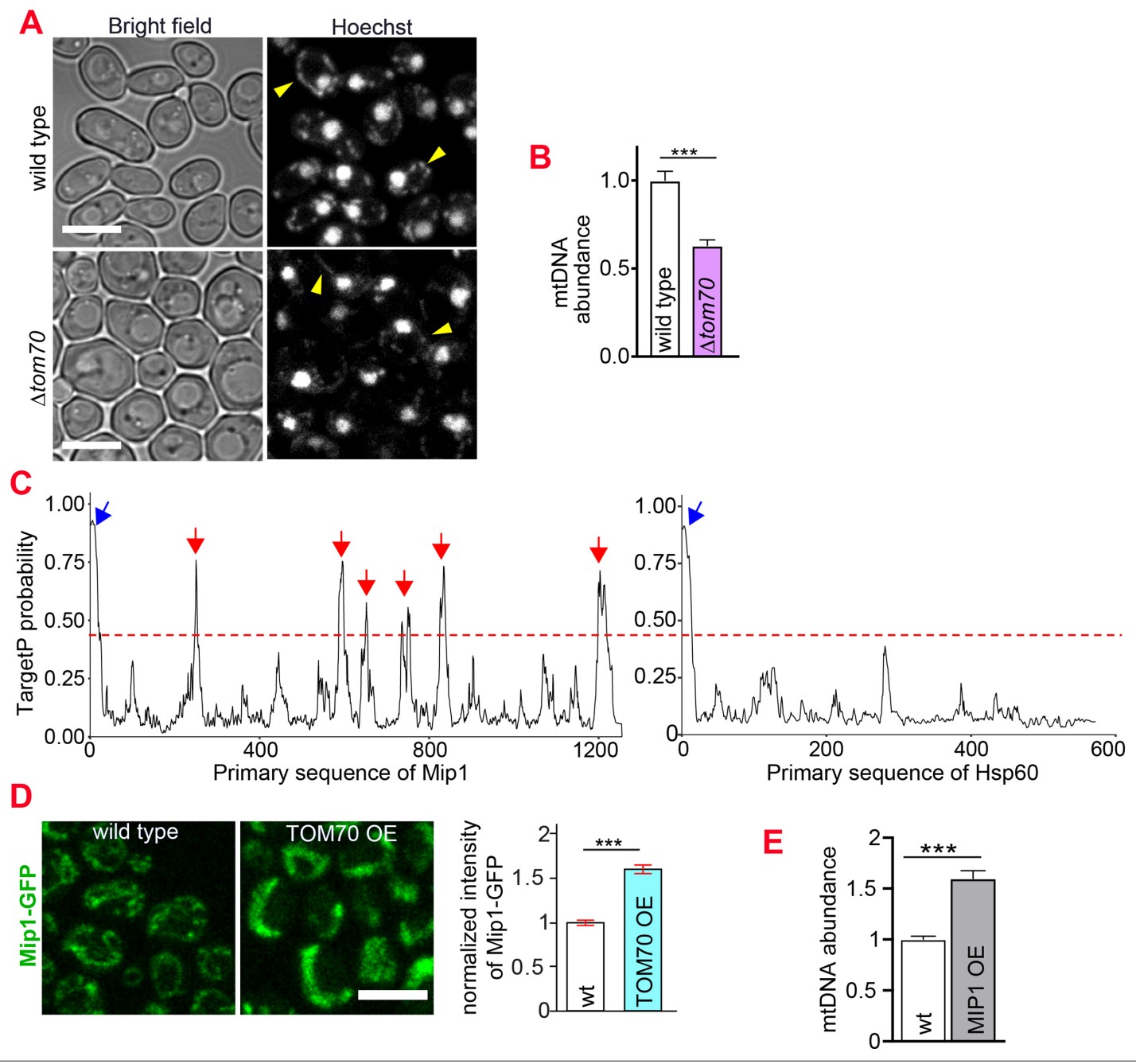

**Figure 3.** Tom70 regulates the abundance of mtDNA. (**A, B**) Representative images and quantification of mtDNA in wild type and *Δtom70* cells were stained by Hoechst dye. Yellow arrowheads point to the mtDNA. Bar graph are the mean and s.e.m. from 87 and 109 cells. (**C**) iMTS probability profiles of Mip1 and Hsp60 predicted by TargetP algorithm. The consecutively N-terminally truncated sequences of the proteins were used as input to calculate the TargetP scores for each residue, which shows internal regions with presequence-like properties, or iMTS. Hsp60, another mitochondrial protein does not depend on Tom70, was also plotted in the same way for comparison (**Backes et al., 2018**). Red line (0.4) indicates the cutoff defined in previous study (**Backes et al., 2018**). Blue and red arrows indicate the N-terminal signal peptide and iMTS, respectively. (**D**) Representative images and quantification of Mip1-GFP in control and TOM70 OE cells. Bar graphs are the mean and s.e.m. from 300 and 438 cells. (**E**) Quantification of mtDNA in control and MIP1 OE cells from Hoechst staining. Bar graphs are the mean and s.e.m. from 253 and 209 cells. Scale bar for all images: 5 μm. Images are representative of at least two independent experiments.Bar graphs are the normalized mean and s.e.m. Data were analyzed with unpaired two-tailed t test: ***, p < 0.001.

The online version of this article includes the following figure supplement(s) for figure 3:

**Figure supplement 1.** The role of mtDNA metabolisms in Tom70's regulatory function of mitochondrial biogenesis.

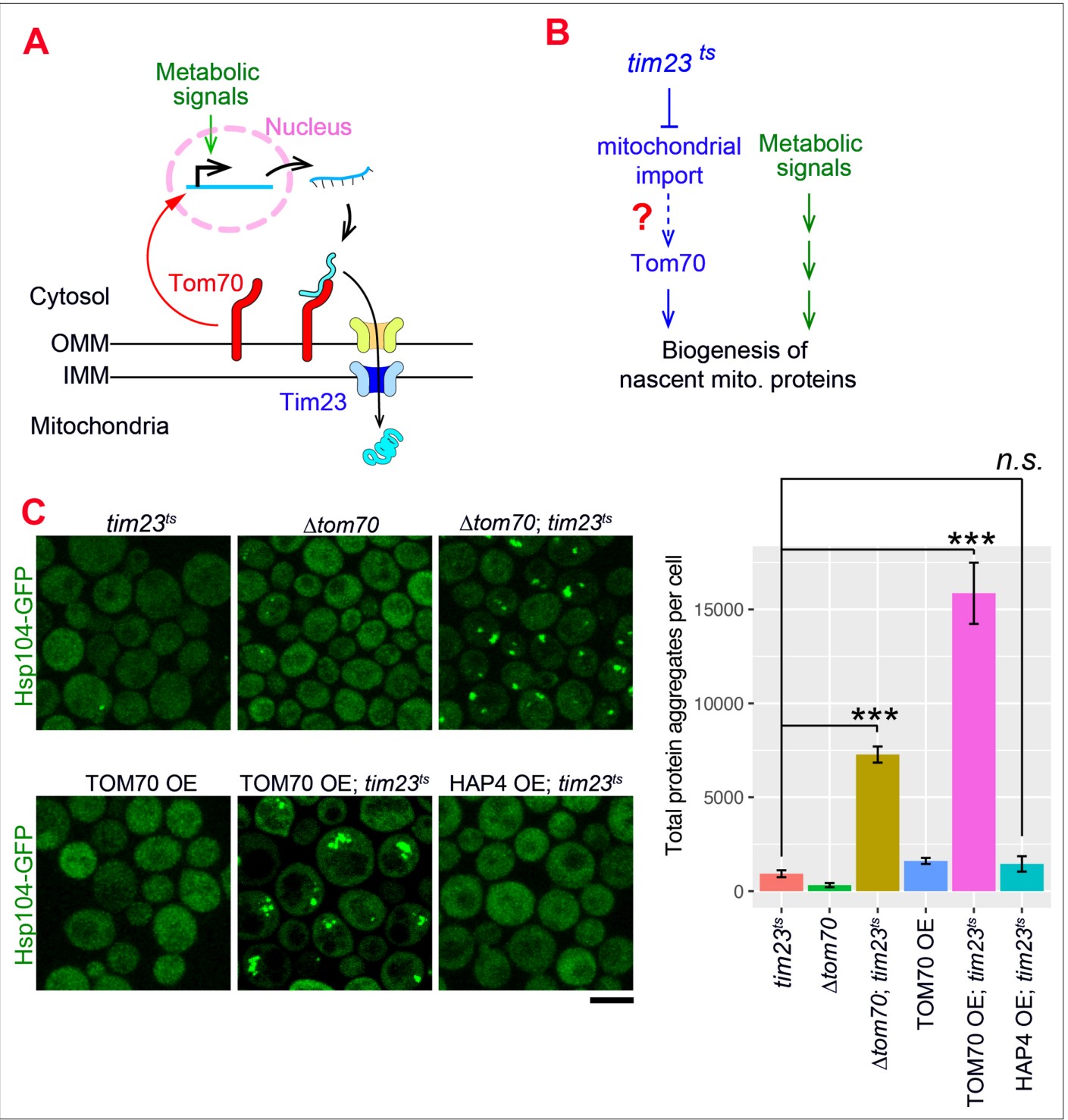

**Figure 4.** The Tom70-dependent regulation of mitochondrial biogenesis is involved in the cellular response to the mitochondrial import defect. (**A**) Mitochondrial biogenesis in wild type cells. Metabolic signals regulate the transcriptional activity of mitochondrial proteins, which are synthesized in the cytosol and translocated through the outer mitochondrial membrane (OMM) and inner mitochondrial membrane (IMM) via TOM-TIM complexes. Tom70, a key component of the TOM complex, serves as the import receptor and the biogenesis regulator of mitochondrial proteins. Tim23, a key component of the TIM complex, acts downstream and receives the incoming nascent mitochondrial proteins from the TOM complex. (**B**) The potential role of Tom70 in the cellular response to the impairment of mitochondrial import caused by *tim23*^ts^. (**C**) Representative images and quantification for different strains after switching to the restricted temperatures to inactivate *tim23*^ts^. All strains were cultured in raffinose medium at 25 °C overnight, followed by adding galactose for 3hrs before adding glucose for 30 min and switching to 35 °C for 2 hr to inactivate *tim23*^ts^. Cytosolic protein

*Figure 4 continued on next page*

*Figure 4 continued*

aggregation was visualized with GFP-tagged Hsp104, a general marker of protein aggregates (*Glover and Lindquist, 1998*; *Zhou et al., 2014*). Bar graphs are the mean and s.e.m. of the Hsp104-GFP signals inside protein aggregates in each cell. Scale bar: 5 µm. Images are representative of at least two independent experiments. Data were analyzed with unpaired two-tailed t test: ***, p < 0.001; n.s., not significant. Sample sizes in (**C**) are given in *Supplementary file 4*.

The online version of this article includes the following figure supplement(s) for figure 4:

**Figure supplement 1.** Model of Tom70's role in the cellular response to the mitochondrial import defect.

## The reduction of Tom70 contributes to the age-related defects of mitochondria

Tom70's transcription regulatory role led us to test whether Tom70 is a key factor in age-related mitochondrial biogenesis defects, a conserved hallmark of aging (*López-Otín et al., 2013*). Previous studies revealed that protein levels were increasingly uncoupled from their transcript levels during aging (*Janssens et al., 2015*). Thus, we chose to follow protein, instead of mRNA, abundance changes during aging to closely reflect the functional and physiological status of the aged cells. We examined different mitochondrial proteins in purified replicative old cells using imaging and APDs, which allowed us to achieve single cell/age resolution, single-photon sensitivity, and avoid the complications caused by the young cells contamination and impurity in the purified old cells that were previously experienced in mass spectrometry-based studies (*Hendrickson et al., 2018*; *Janssens et al., 2015*).

Using this approach, we found that Tom70 underwent age-dependent reduction (*Figure 5A and B*). A similar level of Tom70 reduction was previously reported in aged rats (*Kang et al., 2017*). The reduction of Tom70 is not unique as other mitochondrial proteins, including other TOM proteins, also showed age-dependent reduction (*Figure 5B*), which is consistent with the general trend of mitochondrial biogenesis defect observed in the previous proteomics study of aged cells (*Janssens et al., 2015*). This mitochondria biogenesis defect is associated with the loss of mitochondrial membrane potential, a common hallmark of mitochondrial defects (*Figure 5C*; *Hughes and Gottschling, 2012*; *Sun et al., 2016*). To test if the reduction of Tom70 contributes to the loss of mitochondrial membrane potential in aged cells, we replaced TOM70's promoter with GAL promoter to prevent the age-associated reduction of Tom70. We found that overexpressing TOM70 can rescue mitochondrial membrane potential in aged cells (*Figure 5C and D*). This is unique to Tom70 as overexpressing other TOM proteins cannot fully prevent the loss of mitochondrial membrane potential during aging (*Figure 5D*). As mitochondrial membrane potential plays a key role in mitochondrial biogenesis, pGAL-TOM70 also prevented the age-associated reduction of other mitochondrial proteins (*Figure 5E and F*).

These beneficial effects of TOM70 OE were not caused by the presence of galactose in the medium as both control and TOM70 OE cells, as well as the cells overexpressing other TOM proteins, were cultured in the same medium. In addition, similar results were observed when Tom70 was expressed from the Z3EV promoter in the glucose medium (*Figure 5—figure supplement 1A, B*; *McIsaac et al., 2013*). In contrast, knocking out Tom70 accelerated the loss of mitochondrial membrane potential and biogenesis defects of many other mitochondrial proteins during aging (*Figure 6A–D*, *Figure 6—figure supplement 1*). Moreover, the *Δtom70* cells also lost mtDNA faster than wild-type cells during aging (*Figure 6E*). This is likely due to the accelerated loss of mtDNA polymerase Mip1, which uses Tom70 for import, in *Δtom70* cells during aging (*Figure 6C*). The accelerated loss of mitochondrial biogenesis and functions likely explains previous results that knocking out Tom70 reduces the replicative lifespan (RLS) of yeast (*Schleit et al., 2013*). Consistent with this hypothesis, TOM70 OE, which activates mitochondrial biogenesis, can extend the RLS of budding yeast (*Figure 6F*). This RLS extension is similar to the caloric restriction (CR) of 0.1% glucose (*Kaeberlein et al., 2004*), which is known to increase the expression of many mitochondrial proteins (*Lin et al., 2002*; *Ruetenik and Barrientos, 2015*; *Wuttke et al., 2012*). Therefore, the reduction of Tom70 contributes to the compromised mitochondrial biogenesis in aged cells.

It has been shown that CR can suppress the negative impact of many mitochondrial mutants, including *Δtom70*, on the RLS of yeast (*Schleit et al., 2013*). For example, CR rescues the RLS of *ΔPhb1/2* cells, the mitochondrial prohibitin complex required to stabilize newly imported proteins, by reducing the cytosolic translation which decreases the folding demands of nascent mitochondrial proteins and the load of proteostasis burden inside mitochondria (*Schleit et al., 2013*). As *Δtom70*

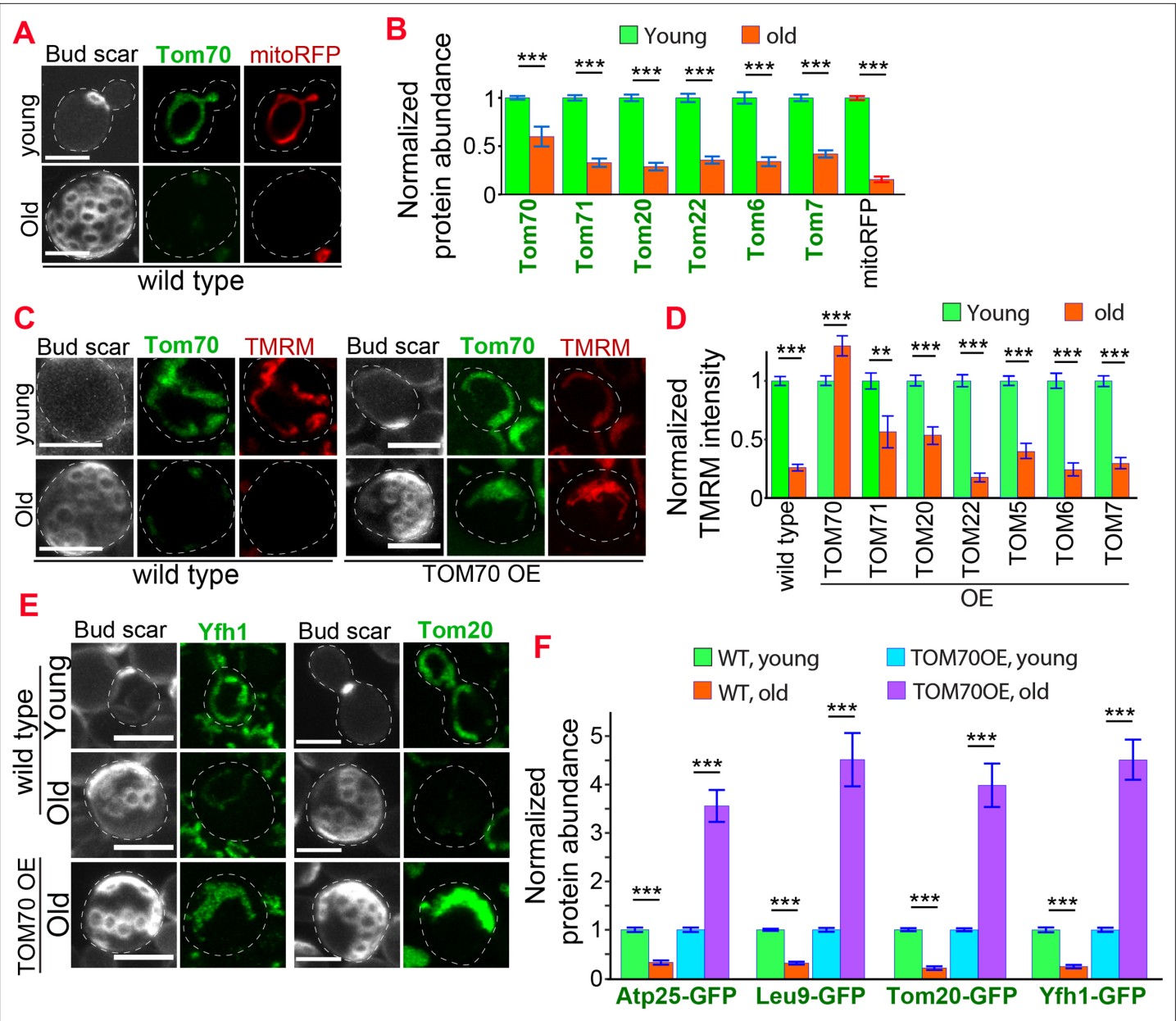

**Figure 5.** Age-related reduction of Tom70 is associated with mitochondrial dysfunctions. (**A**) Representative images of Tom70-GFP and mito-RFP in young and aged cells. Both young and old cells were from YPD culture and stained with calcofluor white to visualize the bud scars. (**B**) Quantification of different GFP-tagged TOM proteins and mitoRFP in young and aged cells. Both young and old cells were from YPD culture and the GFP signal was normalized to corresponding young cells of the same strain. (**C, D**) Representative images (**C**) and quantification (**D**) of mitochondrial membrane potential in young and aged cells from different strains. Different TOM proteins were overexpressed from pGAL promoter. Both young and old cells were from YEP-galactose medium. Mitochondrial membrane potential was indicated by TMRM staining. (**E, F**) Representative images (**E**) and quantification (**F**) of mitochondrial proteins undergo age-associated loss of expression, which can be suppressed by overexpressing TOM70. WT, wild type. Both young and old cells were from YEP-galactose medium. The significant increase of mitochondrial proteins in old TOM70 OE cells (**F**) is likely due to the continuous induction of mitochondrial proteins by TOM70 OE throughout the entire lifespan. Scale bar for all images: 5 µm. Mitochondrial proteins were visualized by endogenous C-terminal GFP tagging and expressed from their own promoters. Both young and old cells in each experiment went through the same culture medium, experimental procedures, and purification. Images are representative of at least two independent experiments. Bar graphs are the mean and s.e.m. that normalized to young cells for each genotype. Sample sizes of (**B, D, F**) are given in *Supplementary file 4*. Data were analyzed with unpaired two-tailed t test: ***, p < 0.001; **, p < 0.01; *, p < 0.05.

The online version of this article includes the following figure supplement(s) for figure 5:

**Figure supplement 1.** Overexpressing Tom70 from a different promoter can rescue age-associated mitochondrial defects.

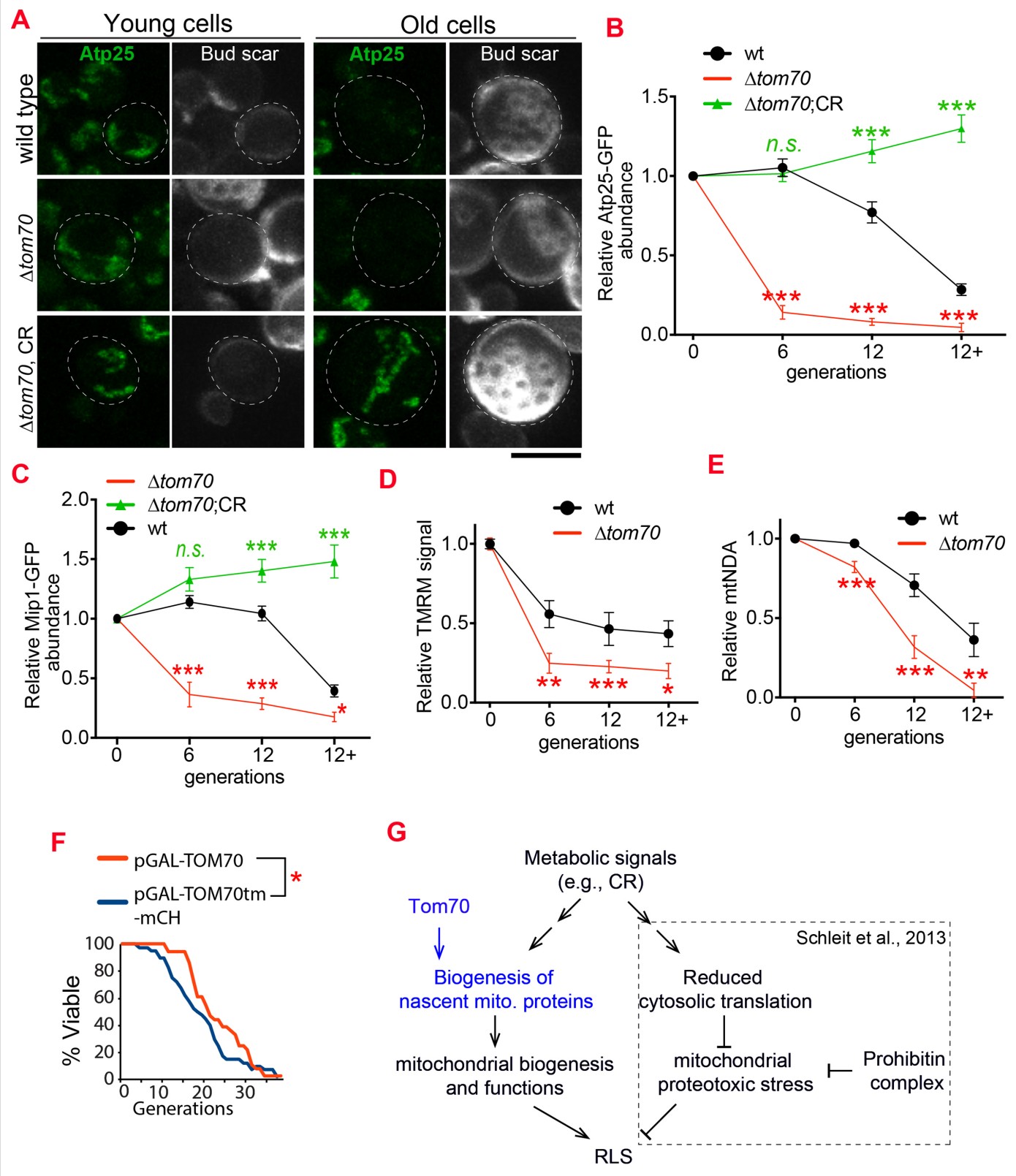

**Figure 6.** Loss of Tom70 accelerates mitochondrial aging. (**A, B**) Representative images (**A**) and quantifications (**B**) of Atp25-GFP in young and aged cells of different strains. Both young and old cells were from YPD culture and stained with calcofluor white to visualize the bud scars. Both young and old cells were quantified from the purified cells that went through the same experimental procedures and purification (same for other results). All strains cultured in YPD medium with 2% glucose except for CR, which has 0.05% glucose. The GFP signal quantified from the old cells was normalized to

*Figure 6 continued on next page*

*Figure 6 continued*

corresponding young cells of the same strain. Old cells were grouped as young (0 bud scar), age 2–6, age 7–11, and age 12+. Data were analyzed with unpaired two-tailed t test: ***, p < 0.001; **, p < 0.01; *, p < 0.05; n.s., not significant (same for C-E). More than 20 cells quantified for each group (same for C-E). (**C**) Quantification of Mip1 in young and aged cells from different strains. All strains cultured in YPD medium with 2% glucose except for CR, which has 0.05% glucose. (**D, E**) Quantification of mitochondrial membrane potential (**D**) and mtDNA (**E**) in young and aged cells from different strains. Mitochondrial membrane potential was indicated by TMRM staining. All strains cultured in YPD medium with 2% glucose except for CR, which has 0.05% glucose. (**F**) Survival curve from replicative life span assay of 36 and 40 cells expressing pGAL-TOM70 and pGAL-TOM70tm-mCH, respectively, were determined by microscopic dissection on YEP-galactose plates. *P* = 0.038 by Mann-Whitney test. (**G**) Proposed model by which CR extends the RLS of different mitochondrial mutants via different mechanisms. *Schleit et al., 2013* showed that CR extends the RLS of Δ*phb* cells by reducing mitochondrial proteotoxic stress via reduced cytosolic translation. Our results demonstrated that this reduced cytosolic translation in CR is not universal as the biogenesis of mitochondrial proteins is preserved during aging. This preserved mitochondrial biogenesis in CR is consistent with the observed rescue of RLS in Δ*tom70* cells by CR (*Schleit et al., 2013*). Scale bar for all images: 5 µm. Mitochondrial proteins were visualized by endogenous C-terminal GFP tagging and expressed from their own promoters. Images are representative of at least two independent experiments.

The online version of this article includes the following figure supplement(s) for figure 6:

**Figure supplement 1.** Additional data for the accelerated loss of mitochondrial proteins in Δ*tom70* cells during aging.

reduces the biogenesis/import of mitochondrial proteins during aging (*Figure 6A–E*), which is different from the mitochondrial proteostasis defect caused by Δ*phb1*, we considered a different mechanism that allows CR to rescue the RLS of Δ*tom70* cells (*Figure 6G*). Indeed, by checking mitochondrial proteins that showed accelerated biogenesis defects during the aging of Δ*tom70* cells, we found that their expression levels were rescued by CR during aging (*Figure 6A–D*, *Figure 6—figure supplement 1*). This suggests that CR restores the RLS of Δ*tom70* cells by preventing their premature mitochondrial biogenesis defect.

## Reduced biogenesis and enhanced degradation underlying the age-dependent reduction of mitochondrial Tom70

Age-related Tom70 reduction is likely a conserved process, as Tom70 protein is also reduced during fly aging and in aged or pathologically hypertrophic hearts from humans and rats (*Kang et al., 2017*; *Li et al., 2014*; *Pacifico et al., 2018*; *Yang et al., 2019*). In addition, overexpression of TOM70 was found to protect the cultured cardiomyocytes against diverse pro-hypertrophic insults (*Kang et al., 2017*; *Li et al., 2014*). Given these important roles of Tom70, we then asked what causes its age-associated reduction. The protein level of Tom70 is determined by its biogenesis and degradation. We first checked the biogenesis of Tom70 during aging. Changing TOM70's promoter prevented its age-dependent reduction, suggesting that the loss of transcriptional activity likely contributes to the reduction of Tom70 during aging (*Figure 5A,B*, vs *Figure 7—figure supplement 1A*). Consistent with this hypothesis, analysis of the published dataset revealed that the mRNA level of TOM70 is reduced in the purified old yeast cells (*Figure 7—figure supplement 1B*; *Janssens et al., 2015*). To find TFs that regulate TOM70 expression, we searched the database for TFs that recognize the promoter of TOM70 (*Monteiro et al., 2020*). We confirmed that these predicted TFs either positively or negatively regulate Tom70-GFP level in young cells (*Figure 7—figure supplement 1C*). Increasing the expression of some, but not all, TFs is able to suppress the age-dependent reduction of Tom70 protein (*Figure 7A*). This suggests that a reduced TF expression level, TF activity, the accessibility of promoters, or any of these factors in combination, contributes to the reduction of Tom70 in aged cells (*Hendrickson et al., 2018*).

In addition to the biogenesis of Tom70, we asked whether aging affects the degradation of Tom70 as well. Previous studies revealed that under acute vacuole deacidification Tom70 can be sorted into mitochondria-derived compartment (MDCs) in a Fis1/Dnm1-dependent pathway for degradation (*Hughes et al., 2016*). We asked whether Fis1 and Dnm1 also regulate Tom70 turnover during physiological aging. Indeed, cells lacking either Dnm1 or Fis1 showed a significantly higher expression level of Tom70 protein in both young (*Figure 7—figure supplement 1D*) and old cells (*Figure 7B and C*). Interestingly, the Dnm1 protein level increases with age (*Figure 7D and E*), which could enhance the degradation of Tom70 and undermine the biogenesis of mitochondria in aged cells.

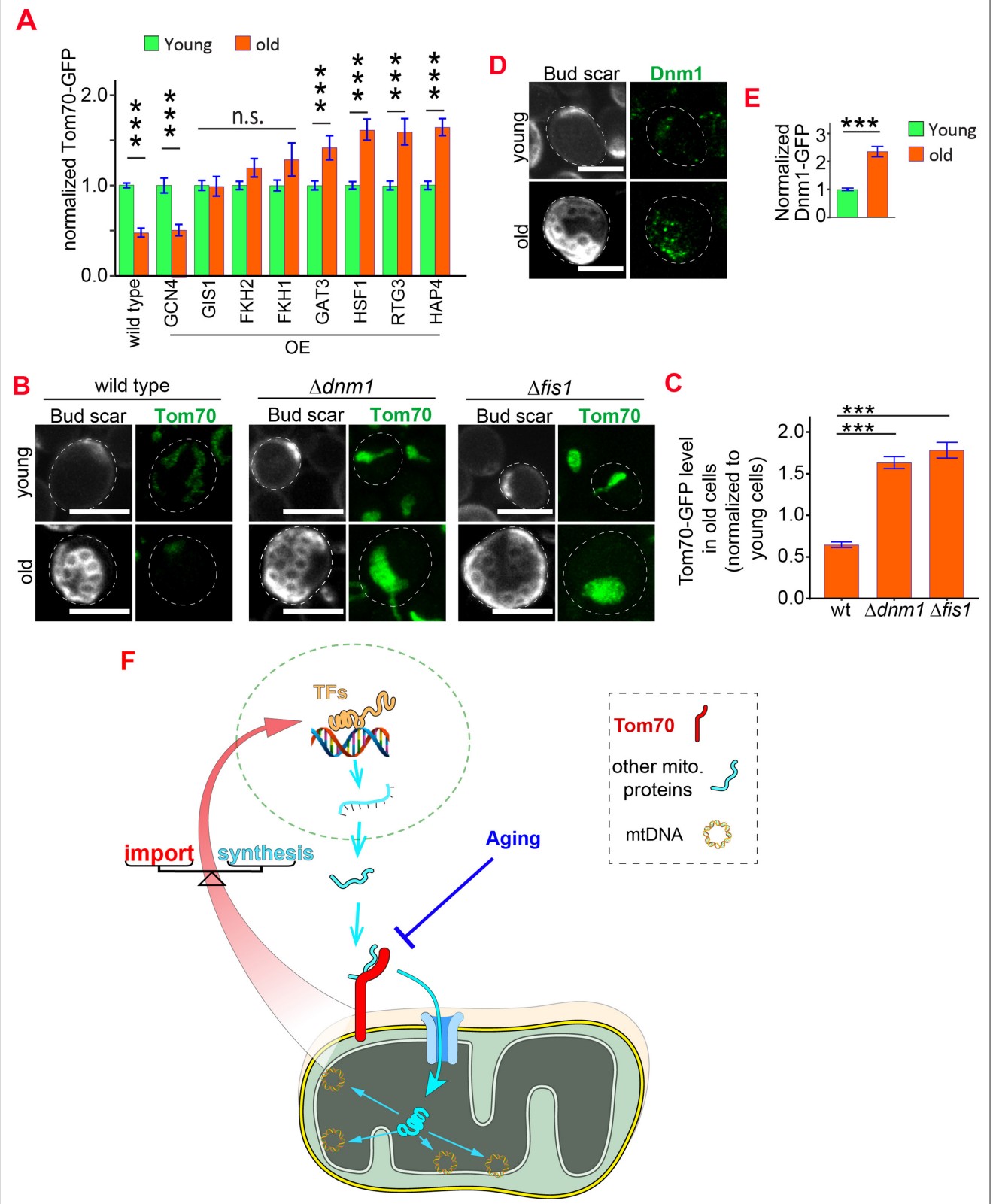

**Figure 7.** Reduced biogenesis and enhanced degradation underlying the age-dependent reduction of mitochondrial Tom70. (**A**) Mean and s.e.m. of Tom70-GFP signal in young and old cells from strains overexpressing different TFs. Sample sizes are given in ***Supplementary file 4***. Both young and old cells were from YEP-galactose medium. (**B, C**) Representative images (**B**) and quantification (**C**) of Tom70-GFP in young and old cells from wild type and mutant strains. Mean and s.e.m. from 57, 154, 50 old cells were normalized to young cells for each strain. Both young and old cells were from YPD

*Figure 7 continued on next page*

*Figure 7 continued*

medium. (**D, E**) Representative images (**D**) and quantification (**E**) of Dnm1-GFP in young and old cells. Mean and s.e.m. from 89 young and 53 old cells were quantified. Both young and old cells were from YPD medium. (**F**) Model of Tom70's roles. Tom70 sits at the crossroad of cytosolic proteostasis and mitochondrial biogenesis by regulating both the biogenesis and import of mitochondrial proteins. The reduction of Tom70 during aging is associated with age-dependent mitochondrial defects. Scale bar for all images: 5 µm. Mitochondrial proteins were visualized by endogenous C-terminal GFP tagging and expressed from their own promoters. Images are representative of at least two independent experiments. Data were analyzed with unpaired two-tailed t test: ***, p < 0.001; **, p < 0.01; *, p < 0.05; n.s., not significant.

The online version of this article includes the following figure supplement(s) for figure 7:

**Figure supplement 1.** Additional data for the mechanisms that control the expression of Tom70 in young and aged cells.

## Discussion

The results described above reveal a previously unknown role of Tom70 in regulating the biogenesis of mitochondrial proteins. Cells use the same molecule, Tom70, to regulate the transcriptional activity and import of mitochondrial proteins. These dual roles of Tom70 connect two main steps of mitochondrial biogenesis that have been studied separately in the past, namely the transcriptional regulations of mitochondrial proteins and the molecular mechanisms of mitochondrial import. The imbalance between import and biogenesis of nascent mitochondrial proteins leads to their cytosolic accumulation, which causes cytosolic proteostasis stress and the formation of cytosolic protein aggregates (*Samluk et al., 2018*; *Topf et al., 2019*; *Wrobel et al., 2015*). We propose that Tom70's dual roles in both biogenesis and import of mitochondrial proteins allow the cells to accomplish mitochondrial biogenesis without compromising cytosolic proteostasis (*Figure 7F*). This interdependence between cytosolic proteostasis and mitochondrial biogenesis is consistent with the observations that mitochondrial dysfunction and protein aggregation are two closely related hallmarks of aging and age-related diseases.

Nascent mitochondrial proteins synthesized in the cytosol are transferred from Hsp70 to Tom70, which then transfers the nascent proteins to Tom40 for translocation into the mitochondria (*Figure 4—figure supplement 1A*). When mitochondrial import is perturbed, such as Tim23 inactivation, the transfer of nascent mitochondrial protein to Tom40 and translocation are halted. These unimported nascent mitochondrial proteins stay on the surface of mitochondria and associate with upstream factors, such as Tom70. Such mitochondrial import defect activates the stress response that represses the biogenesis of mitochondrial proteins in order to relieve the cytosolic accumulation/aggregation of nascent mitochondrial proteins (*Wang and Chen, 2015*; *Wrobel et al., 2015*). Our results show that such repressive program requires Tom70, as the mitochondrial import defect failed to repress the mitochondrial biogenesis and caused cytosolic accumulation/aggregation in the absence of Tom70. Together with the transcriptional activation of mitochondrial biogenesis upon TOM70 OE, these results are consistent with the model that the ratio/balance between Tom70-nascent mitochondrial protein, which represents the capacity of mitochondrial import, regulates the biogenesis activity of mitochondrial proteins (*Figure 4—figure supplement 1*).

Retrograde signaling, such as the one mediated by Rtg1/2/3 in yeast and many forms of mitochondrial unfolded protein response (UPR) (*Shpilka and Haynes, 2018*), has been described for mitochondria to mitigate the proteostasis defect *inside* mitochondria. In contrast, this Tom70-dependent transcriptional regulation of mitochondrial biogenesisis probably evolved to maintain proteostasis *outside* of mitochondria. Consistent with this view, our results showed that this Tom70-nucleus communication is independent of the classic retrograde signaling via Rtg1/2/3 that responds to the mitochondrial defects. Instead, several other TFs are required in this Tom70-dependent transcriptional regulation of mitochondrial biogenesis. Some of these TFs, such as the Forkhead family transcription factor Fkh1/2, could be the missing TFs that mediate the increase of mitochondrial activity in response to certain longevity cues. For example, it was shown that CR activates mitochondria biogenesis independent of Hap4, the well-studied TF of mitochondrial biogenesis in yeast (*Lin et al., 2002*). Mining the published dataset, we found that Fkh1 is increased in CR-treated cells (*Choi et al., 2017*). The orthologs of these Forkhead family transcription factors, such as Daf-16/FOXO, are integrators of different signal pathways to regulate longevity and aging.

Our results also provide a new explanation for the mitochondrial defects in aged cells as the reduction of Tom70 correlates with the loss of mitochondrial membrane potential, many mitochondrial

proteins, and mtDNA, which can be rescued by overexpressing Tom70. Mechanistic dissection of this conserved Tom70 reduction in old cell suggests that age-associated loss of transcriptional activity and increased degradation of Tom70 contribute to its reduction during aging. Many studies showed that loss of proteostasis is an early event during aging (*Ben-Zvi et al., 2009*; *Hipp et al., 2019*; *Yang et al., 2019*). The loss of proteostasis may inactivate some TFs required for the expression of Tom70 and other mitochondrial proteins as many TFs contain low complex domains that are sensitive to proteostasis impairment (*Alberti and Hyman, 2016*; *Boija et al., 2018*; *Liu et al., 2020*; *Vecchi et al., 2020*). In addition, the increased level of Dnm1 during aging could further promote the degradation of mitochondrial Tom70. The reduction of Tom70 and mitochondrial biogenesis could lead to the previously observed age-dependent mitochondrial defects. The fact that loss of Tom70 accelerates aging and age-related mitochondrial defects, while TOM70 OE extends the RLS and delays these mitochondrial dysfunctions, highlights that Tom70 is a key molecule in mitochondrial aging.

## Materials and methods
### Experimental model details

All *S. cerevisiae* strains and plasmids used in this study are listed in the *Supplementary files 1 and 3*. All yeast strains used in this study are based on the BY4741 strain background. Genetic modifications were performed with PCR mediated homologous recombination (*Longtine et al., 1998*) and genotyped with PCR to confirm correct modification and lack of aneuploid for the chromosome that gene of interest is located. The GFP-tagged strains were from the GFP collection (*Huh et al., 2003*). The mutant strains were generated by PCR amplify KANMX cassette from existing mutant strains (Yeast Knockout Collection, GE Dharmacon) or pFA6a-KanMX plasmid to transform By4741 or the strains based on By4741. Expression of proteins from integration plasmids was done by integrating the linearized plasmid into TRP1 locus. For some strains, an empty integration plasmid was first integrated into TRP1 locus to provide the required sequences, such as AMP gene in the backbone of plasmid, for further integration of other plasmids. UAS-TOM70 fly was provided by Dr. Bingwei Lu (*Gehrke et al., 2015*) and crossed with Mef2-GAL4 to get 3rd instar larvae. UAS-TOM70 crossed with w1118 and Mef2-Gal4 crossed with w1118 served as genetic controls.

For confocal imaging, cells were grown at 30 °C in SC glucose medium (790 mg/L of complete supplement mixture (CSM) from Bioworld, 6.7 g/L yeast nitrogen base, 2% glucose) to $OD_{600}$~0.5 and refreshed for an additional 2–3 hr before imaging. For the strains with Gal promoter-driven proteins, cells were grown at 30 °C in SC 2% raffinose medium overnight to $OD_{600}$~0.5 and induced by adding 2% galactose for 5 hr. For qPCR assay, cells were cultured in YPD (2% glucose, 1% yeast extract, 2% peptone), or YEPR (1% yeast extract, 2% peptone, 2% raffinose) medium overnight and then induced with 2% galactose for 5 hr ($OD_{600}$ between 0.3 and 0.4) before harvest. Gal induction was ceased by adding 2% glucose for 30 min before other treatment or imaging. All media was prepared by autoclaving the solution without glucose/raffinose/galactose for 20 min and adding filtered carbon source as indicated.

### Confocal microscopy

Images were acquired using a Carl Zeiss LSM-510 Confocor 3 system with 100 × 1.45 NA Plan-Apochromat objective and a pinhole of one airy unit. The system was driven by Carl Zeiss AIM software for the LSM 510 meta. 405/488/561 nm laser was used to excite Calcofluor-white (CFW)/GFP/RFPs, and emission was collected through the appropriate filters onto the single photon avalanche photodiodes on the Confocor 3. All CFW images were acquired through a 420 nm long pass filter, GFP images were acquired through a 505–540 nm filter, and RFP images were acquired with a 580–610 nm filter on the Confocor 3. All images were acquired in a multi-track, alternating excitation configuration so as to avoid bleed through. GFP-fusion strains were imaged with excitation laser at 8 kW/cm². Any strains used in comparison were acquired with the same laser and scanning setting to compare their expression levels. All image processing was performed in the Image J software (NIH, Bethesda, MD). The total amount of each protein (e.g. Tom70-GFP) or fluorescent dye (e.g. TMRM) was quantified from the sum projection of Z-stacks after extracting the background signals. For visualization purposes, images scaled with bilinear interpolation were used for figures.

## Drug treatment and key reagents

Tetramethylrhodamine, Methyl Ester, Perchlorate (TMRM) (Thermo Fisher Scientific T668) dissolved in DMSO to 1 mM as stock solution, 1:10,000 for 15 min for mitochondrial membrane potential detection and quantification. Calcofluor White Stain (Sigma-Aldrich 18909–100 ML-F) was diluted 200 folds in PBS and incubated with cells for 5 min at room temperature, followed by three washes with PBS. Hoechst 33,342 (Invitrogen H3570) 10 µg/ml was used to stain DNA for 15 min, followed by two washes with fresh medium before imaging. β-estradiol (Sigma E8875) is used at 10 nM or 100 nM as final concentrationto induce Tom70 overexpression from Z3EV promoter.

## RT-qPCR

For RT-qPCR assays, total RNA from yeast cells was isolated with TRIzol LS agent. Each RNA sample (1 µg) was subjected to reverse transcription (Invitrogen SuperScript III One-Step RT-PCR System with Platinum Taq DNA Polymerase, Invitrogen 12574018), and then amplified by real-time PCR (Ssoadvanced Universal SYBR, bio-Rad 1725271). The primers used for qPCR were shown in *Supplementary file 2*. The relative values of gene expression were calculated using the $2^{-\Delta\Delta CT}$ method (*Livak and Schmittgen, 2001*) by comparing the cycle number for each sample with that for the untreated control. The results were normalized to the expression level of tubulin or actin gene. All experiments have three independent replicates and the mean was calculated for the figures. For *Figure 1C*, zy914 and zy930 were induced for 2 hr with 2% galactose to express TEV protease and remove TOM70 cytosolic domain; zy1439, zy1463, zy2974, zy2417, and zy3450 were induced for 5 hr to overexpress Tom70 before harvesting samples for mRNA extraction.

For fly RT-qPCR, total RNA was extracted from 3rd instar larvae (mixed sexes, 4 animals per sample) with the same method described above for yeast. The primers used for fly qPCR were shown in *Supplementary file 5*.

## Isolation of old cells

Yeast cells were collected from a fresh overnight culture and washed twice with cold PBS, pH 8.0. About $4 \times 10^7$ mid-log cells were briefly concentrated by 3,000 g for 20 sec, followed by three washes of cold sterile PBS pH 8.0. Cells were labeled with 1.6 mg/ml EZ-Link Sulfo-NHS-LC-Biotin (Pierce) at room temperature in the same PBS for 30 min with gentle shaking. These cells were used as M-cells. The M-cells were then washed three times with cold PBS, pH 7.2, to get rid of free biotin. These M-cells were resuspended in 500 µl PBS pH7.2 and mixed with 7 µl BioMag magnetic streptavidin beads (Qiagen, 311714) for 60 min at 4 °C and loaded into an acrylic column with curved grade N52 magnet. A continuous flow of fresh medium was applied at 0.3 ml/min to rinse away the daughter cells generated from these M-cells. After 60 hr at room temperature, the aged M-cells were unloaded from the device by removing the magnet and flushed out with 3 ml/min fresh medium. The M-cells were then fixed with 1% formaldehyde for 30 min at room temperature and washed three times with PBS, pH 7.4. After staining with CFW, the cells were imaged with confocal to collect images of the bud scars and also the other channels of interests. For live cell staining (e.g. TMRM staining), the old cells are stained immediately after harvest without fixation. Cells with more than 12 bud scars were treated as old cells.

While developing the old cells purification methodologies in lab, we noticed that environmental stress, such as small amount of NaOH/ethanol left from sterilization step or the formation of crowded cell clusters around the aging cells, caused the accumulation of Tom70-GFP in vesicles for a fraction of old cells. Extensive overnight wash with fresh medium before loading cells into device can eliminate this phenotype. This stress-induced age-dependent accumulation of Tom70-GFP in vesicles for a portion of old cells could explain the phenotypes observed in a recent report using microfluidics (*Li et al., 2020*).

## Lifespan assay

Replicative lifespan were examined as described previously (*Steffen et al., 2009*). Fresh YEP-2% galactose (YEPG) plates were prepared 1 day before the assay. Fresh colonies from different strains were used to setup overnight culture in YPED and then spread 5 µl onto one side of the YEPG plates. Individual cells were aligned in the middle of plates with 20 cells per strain per plate. Both strains were included side-by-side with each other on each plate to reduce variations among plates. The

plates were stored at 4 °C each night and kept at 30 °C during the day. Each day the plates were kept at 30 °C for about 10 hr during the dissections. The experiments were performed blindly by experimenter with the strain names and genotypes encoded by numbers.

## Statistical analysis

All experiments were repeated multiple times to confirm reproducibility. Data are representative of at least three independent experiments. All quantifications are presented as the means ± standard error of mean (s.e.m.). Statistical test for each bar graph in figures was determined by unpaired t test with Welch's correction in Prism 9. n.s. or ns, not significant; $*p < 0.05$; $**p < 0.01$; $***p < 0.001$. Mann-Whitney test was used for the RLS experiment.

## Acknowledgements

The authors thank E Verdin, G Lithgow, and P Walter for discussion. This work was supported by Glenn postdoctoral fellowship to Q Liu, R01 AG058742 to BK Kennedy, and DP5OD024598 and R03AG070478 to C Zhou.

## Additional information

### Funding

| Funder | Grant reference number | Author |
|---|---|---|
| Glenn Foundation for Medical Research | Glenn postdoctoral fellowship | Qingqing Liu |
| NIH Office of the Director | DP5OD024598 | Chuankai Zhou |
| National Institute on Aging | R01 AG058742 | Brian K Kennedy |
| National Institute on Aging | R03AG070478 | Chuankai Zhou |

The funders had no role in study design, data collection and interpretation, or the decision to submit the work for publication.

### Author contributions

Qingqing Liu, Data curation, Formal analysis, Funding acquisition, Investigation, Methodology, Resources, Validation, Visualization, Writing – original draft, Writing – review and editing; Catherine E Chang, Data curation, Investigation, Methodology, Writing – review and editing; Alexandra C Wooldredge, Data curation, Investigation, Writing – review and editing; Benjamin Fong, Formal analysis, Software, Writing – review and editing; Brian K Kennedy, Funding acquisition, Resources, Writing – review and editing; Chuankai Zhou, Conceptualization, Formal analysis, Funding acquisition, Investigation, Methodology, Project administration, Resources, Software, Supervision, Validation, Visualization, Writing – original draft, Writing – review and editing

### Author ORCIDs

Chuankai Zhou (iD) http://orcid.org/0000-0002-0739-0350

### Decision letter and Author response

Decision letter https://doi.org/10.7554/eLife.75658.sa1
Author response https://doi.org/10.7554/eLife.75658.sa2

## Additional files

### Supplementary files

- Supplementary file 1. Experimental models, organisms and strains.
- Supplementary file 2. Oligo DNA.
- Supplementary file 3. Recombinant DNA.

- Supplementary file 4. Source data for sample sizes related to several figures.
- Supplementary file 5. Oligo DNA for fly genes tested.
- Transparent reporting form

## Data availability

The published article includes all datasets generated or analyzed during this study. All original raw data can be accessed in Dryad Digital Repository, doi:https://doi.org/10.5061/dryad.d7wm37q2n.

The following dataset was generated:

| Author(s) | Year | Dataset title | Dataset URL | Database and Identifier |
|-----------|------|---------------|-------------|-------------------------|
| Zhou C | 2022 | Tom70-based transcriptional regulation of mitochondrial biogenesis and aging | https://dx.doi.org/10.5061/dryad.d7wm37q2n | Dryad Digital Repository, 10.5061/dryad.d7wm37q2n |

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
