## [Editor Report]

The authors test the hypothesis that components of the TOM complex regulate efficient mitochondrial biogenesis by coordinating the synthesis (via controlling transcription of the corresponding RNAs) of mitochondrial proteins with the rate of mitochondrial protein import. It has previously been established that failure to import mitochondrial proteins results in the accumulation of toxic protein aggregates in the cytosol. The authors conclude that Tom70 fulfills this role and find that Tom70 expression declines as cells age, which contributes to age-associated mitochondrial dysfunction.

---

## [Decision Letter]

**Decision letter after peer review:**

Thank you for submitting your article "Tom70-based transcriptional regulation of mitochondrial biogenesis and aging" for consideration by *eLife*. Your article has been reviewed by 3 peer reviewers, including Martin Sebastian Denzel as Reviewing Editor and Reviewer #1, and the evaluation has been overseen by Carlos Isales as the Senior Editor.

Essential revisions:

The reviewers agree that the paper is of great interest and there is also agreement on some open questions that would need to be addressed for publication at *eLife*.

1. How does Tom70 influence the synthesis of mitochondrial components? What TFs might be involved and how does this affect the basic mitochondrial transcription machinery?

2. Improvements in the last section are needed. The link between protein homeostasis and the function of LCD-containing TFs in Tom70 expression is not fully developed.

*Reviewer #1 (Recommendations for the authors):*

To improve the paper, the authors should address some of the points below:

In WT, Tom70 seems to be rate-limiting for transcription of mito transcripts. Is it also limiting for mitochondrial import or are other components of the import machinery more critical?

The manuscript suggests that Tom70 coordinates the synthesis and import of mitochondrial proteins. What happens if import is perturbed independently from Tom70 function? Then, Tom70 should still promote transcription/biogenesis, leading to an imbalance of synthesis and import.

Tom70 OE induces expression of nuclear encoded mitochondrial genes. How specific is this? Does Tom70 OE also induce the transcription of non-mito genes? RNA seq might give a fuller picture here.

The data in Figure 4 suggest that Tom70 acts to repress mitochondrial protein biogenesis, while the data in the previous figures suggest that Tom70 acts to promote it (this is then also suggested by the aggregation observed in the OE). This is confusing and needs a better explanation.

*Reviewer #2 (Recommendations for the authors):*

The major weakness relates to the underdeveloped approaches and the vagaries related the mechanism(s) by which Tom70 influences transcription of mitochondrial components. Additional insight into how Tom70 coordinates transcription as a function of mitochondrial protein import "rate/flux" is required to substantiate the author's claims.

In addition to quantifying mtDNA via imaging, relative quantities should be determined by qPCR.

*Reviewer #3 (Recommendations for the authors):*

1. Several proteins are increased to ~1.5 fold (Figure 1B). How significant are the increase in these protein levels? Can this be verified with other standard techniques like western blots? Towards the end, the authors also explore the effect of proteostasis defect on TOM70 expression and mitochondrial biogenesis. This section seems like an off shoot of this study. Further exploration is required for this.

2. With the overexpression of TOM proteins and Hap4, is the level of expression of basic mitochondrial transcription machinery like POLRMT, TFB2M and TFAM increased? Can this in turn have an indirect effect on the observed increase in other protein levels?

3. Figure 1B should include all the proteins shown in Figure 1A like Tom5 and Tom6 although their expression levels did not change. Please also include quantitation of Hap4 overexpression.

4. In Figure 1C, please mark the proteins that are not TOM70's substrates.

5. From Figure 1C, the authors conclude that 'Tom70 levels on the mitochondrial outer membrane control the transcriptional activity of many mitochondrial proteins.' This reviewer is not convinced that the cytosolic domain has no contribution. The conclusion might be a bit more nuanced here than presented.

6. The effect of proteostasis defect on TOM70 expression and mitochondrial biogenesis is a little unclear. How pervasive is the effect in context of LCD-containing TFs? This reviewer feels that this section should either be removed and included in a different study or explored further to be made more relevant here.

---

## [Author Response]

Essential revisions:The reviewers agree that the paper is of great interest and there is also agreement on some open questions that would need to be addressed for publication at eLife.1. How does Tom70 influence the synthesis of mitochondrial components? What TFs might be involved and how does this affect the basic mitochondrial transcription machinery?

We have identified the downstream TFs that respond to Tom70 level and demonstrated that several TFs are required for Tom70 to regulate the biogenesis of mitochondrial proteins (Figure 2). As we described in the manuscript, we think multiple TFs receive input from Tom70 (line 184-188).

We also tested how Tom70 affects the basic mitochondrial transcription machineries. Similar to other mitochondrial proteins we tested, Tom70 OE can transcriptionally up-regulate the yeast homologs of POLRMT, TFB2M and TFAM (Rpo41, Mtf1, and Abf2, respectively) (Author response image 1). However, up-regulating individual mitochondrial transcription machinery proteins did not have much effect in the biogenesis of other mitochondrial proteins compared to TOM70 OE (Author response image 1), indicating that they are not the key mechanism behind Tom70’s effect. In fact, the overexpression of individual mitochondrial transcription machinery proteins causes reduction of other mitochondria proteins, consistent with other data in Figure 1 that Tom70 has a unique function in regulating mitochondrial biogenesis. Although these mitochondrial transcription machinery proteins are likely not important in Tom70-mediated biogenesis, we found that the mitochondrial translation is partially required to increase the biogenesis of some mitochondrial proteins in Tom70 OE (Author response image 1) . We have added these new data as Figure 3—figure supplement 1 and discussed in line 216-222.

**Author response image 1. sa2fig1:** The role of mtDNA metabolisms in Tom70’s regulatory function of mitochondrial biogenesis. (A) mRNA abundance of basic mitochondrial transcription machineries quantified by qPCR in TOM70 OE strain and normalized to control cells. All different yeast strains, including wild type control, were cultured in the same medium. Shown are the mean and s.e.m. from three replicates. (B, C) Quantification (B) of different mitochondrial proteins in wild type control strain and other strains overexpressing different mitochondrial transcription machineries. All different yeast strains were cultured in the same medium. Only RPO41 OE increased Tim44-GFP expression among these strains. In fact, most of these overexpressing strains reduced the mitochondrial abundance, and the mitochondrial morphology in RPO41 OE cells (C) is different from TOM70 OE (Figure 1A), suggesting that these mitochondrial transcription machineries are not downstream of TOM70 OE. (D) Quantification of different mitochondrial proteins in wild type control strain and TOM70 OE strains with/without Erythromycin (2mg/ml) treatment. All different yeast strains were cultured in the same growth medium. Inhibition of mitochondrial translation by Erythromycin partially inhibit the mitochondrial biogenesis induced by TOM70 OE. Data were analyzed with unpaired two-tailed t test: ***, p<0.001; *, p<0.05; n.s., not significant.

In addition, we tried several potential mechanisms based on literature, such as the classical retrograde signaling pathway Rtg1/2/3 (Figure 1D and Figure 1—figure supplement 1E, F). We also tested whether reactive oxygen species (ROS) could mediate the downstream signaling of TOM70 OE as low levels of ROS were proposed to have important roles in mitochondrial retrograde signaling and mitochondrial hormesis (Kim and Koh, 2017; Ristow and Schmeisser, 2014). Similar to Fkh1/2 in Figure 2E, treating cells with ROS scavenger N-acetylcysteine (NAC) partially affected the mitochondrial biogenesis induction in TOM70 OE cells (Author response image 2). The result has been added in line 191.

**Author response image 2. sa2fig2:** Quantification of different mitochondrial proteins in wild type control strain and TOM70 OE strains with/without NAC (20mM) treatment. All different yeast strains were cultured in the same growth medium. ROS scavenger NAC partially inhibits the induction of some mitochondrial proteins (highlighted as yellow bars) in TOM70 OE cells. Data were analyzed with unpaired two-tailed t test: ***, p<0.001; *, p<0.05; n.s., not significant.

All these data are consistent with the model that Tom70 regulates the mitochondrial biogenesis via multiple pathways as we found in our TFs screen (multiple TFs respond to Tom70) (Figure 2) (line 184-188). Disrupting one factor at a time only affects a specific pathway and TF, and therefore, has a limited effect on some, but not all, mitochondrial proteins (Figure 2E). As we described in the manuscript, each TF is responsible for regulating a subset of the mitochondrial proteome. For example, the well-studied Hap4 regulates ~1,089 genes, but only ~237 of them are mitochondrial proteins (which by itself consists of >1,000 proteins) and the other ~ 852 genes are related to Golgi, cytosolic ribosomes, cell wall etc. (Monteiro et al., 2020). After trying many hypotheses, we think it will be a long-term goal for us to fill the missing steps between Tom70 and these multiple identified TFs, such as an unbiased screen for the secondary messengers that relay the signals from Tom70 to these TFs. This will be beyond the scope of current manuscript and we think it is important to share our current discoveries with the field and allow researchers to connect Tom70 to their favorite signaling pathways.

2. Improvements in the last section are needed. The link between protein homeostasis and the function of LCD-containing TFs in Tom70 expression is not fully developed.

We agree that the LCD-proteostasis defect part is not fully developed and we can remove this result according to Reviewer#3’s suggestion.

Reviewer #1 (Recommendations for the authors):To improve the paper, the authors should address some of the points below:In WT, Tom70 seems to be rate-limiting for transcription of mito transcripts. Is it also limiting for mitochondrial import or are other components of the import machinery more critical?

Tom70 also affects mitochondrial import rate as shown by many other papers we cited. It is a general belief that Tom70 is critical for the import of mitochondrial proteins that do not contain typical signal peptides. This description can be found in our manuscript between line 76-89.

The manuscript suggests that Tom70 coordinates the synthesis and import of mitochondrial proteins. What happens if import is perturbed independently from Tom70 function? Then, Tom70 should still promote transcription/biogenesis, leading to an imbalance of synthesis and import.

Nascent mitochondrial proteins synthesized in the cytosol are transferred from Hsp70 to Tom70, which then transfers the nascent proteins to Tom40 and Tim23 for translocation into the mitochondria. When mitochondrial import is perturbed independently of Tom70, such as Tim23 inactivation (Author response image 3), the transfer of nascent mitochondrial protein to Tom40 and translocation is stopped. These unimported nascent mitochondrial proteins stay on the surface of mitochondria and associate with upstream factors, such as Tom70. This mitochondrial import defect, such as the one caused by Tim23 inactivation, is previously known to activate a stress response that represses the biogenesis of mitochondrial proteins (Wang and Chen, 2015; Wrobel et al., 2015). This stress response allows the cell to rebalance the biogenesis with import and prevents cytosolic accumulation/aggregation of nascent mitochondrial proteins. Our results showed that Tom70 is required for such stress response: in the absence of Tom70, Tim23-inactivated cells could not rebalance the synthesis with impaired mitochondria import and cause cytosolic protein aggregation (Figure 4). Therefore, we think the ratio/balance between Tom70-nascent mitochondrial protein, which represents the capacity of mitochondrial import, regulates the biogenesis activity of mitochondrial proteins (summarized in Figure 4A and Figure 4—figure supplement 1). Consistently, when Tom70 is overexpressed in the presence of Tim23 inactivation, there is more Tom70 than nascent mitochondrial proteins, so signals are sent to the nucleus to increase the biogenesis of mitochondrial proteins. This causes an imbalance between biogenesis and import resulting in protein aggregation in the cytosol (Figure 4C and Figure 4—figure supplement 1). Detailed description can be found in our manuscript between line 226-260.

**Author response image 3. sa2fig3:** Schematic of mitochondrial import steps involve Hsp70 and Tom70. In yeast, Hsp70 binds and maintains nascent mitochondrial proteins in unfolded states before transferring them to Tom70 for import (Young et al., 2003). Tim23, the key subunits of the inner membrane translocase TIM complex, acts downstream of TOM complex (including the subunit Tom70) in mitochondrial import. Tim23 inactivation blocks the mitochondrial import of nascent mitochondrial proteins, which stay on the surface of mitochondria and associate with upstream factors, such as Tom70.

Tom70 OE induces expression of nuclear encoded mitochondrial genes. How specific is this? Does Tom70 OE also induce the transcription of non-mito genes? RNA seq might give a fuller picture here.

We do not know if Tom70 OE only specifically regulates mitochondrial proteins, but it will not be surprising if some other genes not related to mitochondria also respond to Tom70: it is known that most of the well-studied mitochondrial biogenesis programs/mechanisms are not specific to the mitochondrial proteome. For example, the well-studied Hap4 regulates ~1,089 genes, but only ~237 of them are mitochondrial proteins (which by itself consists of >1,000 proteins) and the other ~ 852 genes are related to Golgi, cytosolic ribosomes, cell wall etc. (Monteiro et al., 2020). This is because most of the TF-regulated biogenesis programs are integrated into other cellular metabolisms in order to orchestrate different cellular functions. Similarly, these TFs (e.g., Fkh1/2 and Hap4) that responded to Tom70 are known to regulate multiple different aspects of cell biology. Therefore, it is expected that some other genes will be co-regulated along with the mitochondrial proteins studied here. The authors appreciate this question but feel it does not affect the current focus of our story.

The data in Figure 4 suggest that Tom70 acts to repress mitochondrial protein biogenesis, while the data in the previous figures suggest that Tom70 acts to promote it (this is then also suggested by the aggregation observed in the OE). This is confusing and needs a better explanation.

Our response to the second question “What happens if import is perturbed independently from Tom70 function?” also addresses this question. We think the ratio/balance between Tom70-nascent mitochondrial protein, which represents the capacity of mitochondrial import, regulates the biogenesis activity of mitochondrial proteins (summarized in Figure 4A and Figure 4—figure supplement 1). We have added additional discussion between line 387-401.

Reviewer #2 (Recommendations for the authors):The major weakness relates to the underdeveloped approaches and the vagaries related the mechanism(s) by which Tom70 influences transcription of mitochondrial components.

We have identified the downstream TFs that respond to Tom70 level and demonstrated that several TFs are required for Tom70 to regulate the biogenesis of some mitochondrial proteins (Figure 2). As we described in the manuscript, we think multiple TFs receive input from Tom70.

We tried several potential mechanisms based on literature (see discussion in the “Essential Revision”). Our results in this revision and the original data in Figure 2 suggest that Tom70 regulates the mitochondrial biogenesis via multiple pathways as we found in our TFs screen (Figure 2). Disrupting one factor at a time only affects specific pathway and TF, therefore, have limited effect on some, but not all, mitochondrial proteins (e.g., Figure 2E). As we described in the manuscript, each TF is responsible for regulating part of the mitochondrial proteome. For example, the well-studied Hap4 regulates ~1,089 genes, but only ~237 of them are mitochondrial proteins (which by itself consists of >1,000 proteins) and the other ~ 852 genes are related to Golgi, cytosolic ribosomes, cell wall etc. (Monteiro et al., 2020). The broad regulatory effect of Tom70 in mitochondrial biogenesis is consistent with a model that while each metabolic signal regulates a unique subset of mitochondrial proteomes, Tom70 sets the bandwidth for the expression levels of these mitochondrial proteins activated by metabolic signals due to Tom70’s role in mitochondrial import (see model in Figure 4A). This is probably achieved by sending signals to multiple TFs (e.g., Fkh1/2, Hap4, Snf1 in Figure 2A) to finetune the expression level of most mitochondrial proteins.

Additional insight into how Tom70 coordinates transcription as a function of mitochondrial protein import "rate/flux" is required to substantiate the author's claims.

The role of Tom70 in mitochondrial import is well-studied. The main contribution of our results is the discovery of Tom70’s moonlighting role in regulating the transcriptional activity (biosynthesis) of mitochondrial proteins. Collectively our data show that the amount of Tom70 on mitochondrial surface signals to nucleus to regulate the biogenesis activity of mitochondrial proteins: overexpression of TOM70 increases the amount of Tom70 on mitochondrial membrane, which shifts the balance between Tom70-substrates (nascent mitochondrial proteins) and induces the transcriptional activity of mitochondrial proteins in the nucleus (Figure 4—figure supplement 1). We also showed that TOM70 OE increases the amount of other TOM subunits (Figure 1C). This increases the mitochondrial import capacity to match the increased biosynthesis of nascent mitochondrial protein, explaining the lack of protein aggregation in TOM70 OE cells.

In the case of reduced mitochondrial import rate, such as in Tim23 inactivation, the unimported nascent mitochondrial proteins accumulate on mitochondrial surface and Tom70. This is because nascent mitochondrial proteins synthesized in the cytosol are transferred from Hsp70 to Tom70, which then transfers the nascent proteins to Tom40 and Tim23 for translocation into mitochondria (Author response image 3). When mitochondrial import is perturbed in Tim23 inactivation, the transfer of nascent mitochondrial protein to Tom40 and the downstream translocation is stopped, and these unimported nascent mitochondrial proteins stay on the surface of mitochondria and associate with upstream factors, such as Tom70. Previous studies have shown that these unimported nascent mitochondrial proteins cause the stress response that represses the biogenesis of mitochondrial proteins in order to relieve the cytosolic accumulation/aggregation of nascent mitochondrial proteins (Wang and Chen, 2015; Wrobel et al., 2015). Our results show that this repressive program requires Tom70, as the mitochondrial import defect failed to repress the mitochondrial biogenesis and caused cytosolic accumulation/aggregation in the absence of Tom70 (Figure 4C). Detailed description can be found in our manuscript between line 226-260 and Figure 4—figure supplement 1.

In addition to quantifying mtDNA via imaging, relative quantities should be determined by qPCR.

Quantitative imaging approach is an established method to measure DNA (Gomes et al., 2018). In fact, our single-photon sensitive APD brings this quantification to a new level of sensitivity and accuracy with single cell resolution. The basic principle behind quantitative imaging and qPCR is similar: detect the amount of fluorescent dye that bind the DNA. The difference is that qPCR requires purification of DNA from millions of cells and many runs of amplification before the detector can quantify the signal. The single photon counting detector (APD) in our quantitative imaging removes the requirement of DNA purification and amplification steps and directly counts the DNA content *in situ*.

Reviewer #3 (Recommendations for the authors):1. Several proteins are increased to ~1.5 fold (Figure 1B). How significant are the increase in these protein levels?

The statistical analysis is given in the result already and color-coded in Figure 1B. The ~1.5 fold induction of some proteins are statistically significant and comparable to the well-studied HAP4 OE results both here and published by other groups (Figure 1A, Figure 1—figure supplement 1A) (Lascaris et al., 2003). For example, a previous study using spectral absorption analysis found that the cytochrome c and c1 increased by ~1.66 times, cyto b increased ~1.51 times, upon HAP4 OE (Lascaris et al., 2003). In fact, most literature on mitochondrial biogenesis measured the mRNA abundance of mitochondrial proteins. The increase of mRNA abundance upon TOM70 OE in Figure 1C is comparable or higher than the cells with HAP4 OE published by other groups (Lin et al., 2002). The color scale of heatmap in Figure 1C may cause underestimation of the mRNA fold increase in TOM70 OE cells. We annotated the fold change on top of the current heatmap to clarify this.

Can this be verified with other standard techniques like western blots?

As explained at the beginning of our *Result* section, the quantitative imaging method we used here is capable of detecting single GFP with single cell resolution. In contrast, traditional Western blot requires protein extraction from millions of cells and use regular CCD detector (e.g., ChemiDoc Imaging Systems from Bio-Rad Laboratories), which is much less sensitive. Therefore, compared to the basic techniques such as Western blots, the single photon counting detector (APD) allowed us to see statistically significant expression difference *in situ* even for proteins with limited fold changes. In response to reviewer’s request, we randomly chose a few proteins from the list that showed ~1.5-fold changes (Put2, Yta10, Ahd3) and ran WB for them. Compared to Tim23 and Tom7, which showed >1.5-fold increase in Figure 1B, we can see the difference, although not dramatic due to the sensitivity of WB (Author response image 4) .

**Author response image 4. sa2fig4:** Representative WB of WT and TOM70 OE strains.

Towards the end, the authors also explore the effect of proteostasis defect on TOM70 expression and mitochondrial biogenesis. This section seems like an off shoot of this study. Further exploration is required for this.

We agree that it is unclear regarding the mechanistic details of how proteostasis defects contribute to the reduction of Tom70 during aging. This limitation has been stated in the manuscript (line 334-336 of the original manuscript). However, we think this is not a critical point in our manuscript and we removed this section in the revised manuscript.

2. With the overexpression of TOM proteins and Hap4, is the level of expression of basic mitochondrial transcription machinery like POLRMT, TFB2M and TFAM increased?

Indeed, these basic mitochondrial transcriptional machineries are increased in TOM70 OE cells (Author response image 1). Here, the yeast homologs for POLRMT, TFB2M, and TFAM are Rpo41, Mtf1, and Abf2.

Can this in turn have an indirect effect on the observed increase in other protein levels?

We have tried OE of these proteins individually or in combination for Mtf1&Rpo41 (Mtf1 interacts with and stabilizes Rpo41 on promoter), but most of them were not able to mimic the effect of Tom70 OE. These results further strengthen the unique role of Tom70 in regulating mitochondrial biogenesis.

3. Figure 1B should include all the proteins shown in Figure 1A like Tom5 and Tom6 although their expression levels did not change. Please also include quantitation of Hap4 overexpression.

Quantification was added as Figure 1-supplement 1A as suggested.

4. In Figure 1C, please mark the proteins that are not TOM70's substrates.

According to literature, we marked the validated Tom70 substrates in red.

5. From Figure 1C, the authors conclude that 'Tom70 levels on the mitochondrial outer membrane control the transcriptional activity of many mitochondrial proteins.' This reviewer is not convinced that the cytosolic domain has no contribution. The conclusion might be a bit more nuanced here than presented.

Tom70 has most of its molecular mass facing the cytosol (aa 39-617, or cytosolic domain in the manuscript). When we express Tom70’s cytosolic domain in the cytosol, we see very limited induction of mitochondrial biogenesis (Figure 1C) compared to the overexpression of full length Tom70 which is anchored on mitochondrial outer membrane. We think the cytosolic domain of Tom70 must be anchored on the mitochondrial in order to regulate the biogenesis of mitochondrial proteins. Therefore, our original description does not conclude that cytosolic domain has no contribution (line 159-162). In fact, we concluded that Tom70 has to be on the mitochondrial membrane in order to regulate the mitochondrial biogenesis program. We apologize for not making this point clear. We have modified the related text to clarify this point further in line 159.

6. The effect of proteostasis defect on TOM70 expression and mitochondrial biogenesis is a little unclear. How pervasive is the effect in context of LCD-containing TFs? This reviewer feels that this section should either be removed and included in a different study or explored further to be made more relevant here.

We removed this section.

Bibliography

Gomes, C.J., Harman, M.W., Centuori, S.M., Wolgemuth, C.W., Martinez, J.D., 2018. Measuring DNA content in live cells by fluorescence microscopy. Cell Div. 13, 6. doi:10.1186/s13008-018-0039-z

Hughes, A.L., Hughes, C.E., Henderson, K.A., Yazvenko, N., Gottschling, D.E., 2016. Selective sorting and destruction of mitochondrial membrane proteins in aged yeast. *eLife* 5. doi:10.7554/*eLife*.13943

Kim, S., Koh, H., 2017. Role of FOXO transcription factors in crosstalk between mitochondria and the nucleus. J Bioenerg Biomembr 49, 335–341. doi:10.1007/s10863-017-9705-0

Lascaris, R., Bussemaker, H.J., Boorsma, A., Piper, M., van der Spek, H., Grivell, L., Blom, J., 2003. Hap4p overexpression in glucose-grown *Saccharomyces cerevisiae* induces cells to enter a novel metabolic state. Genome Biol. 4, R3. doi:10.1186/gb-2002-4-1-r3

Lin, S.-J., Kaeberlein, M., Andalis, A.A., Sturtz, L.A., Defossez, P.-A., Culotta, V.C., Fink, G.R., Guarente, L., 2002. Calorie restriction extends *Saccharomyces cerevisiae* lifespan by increasing respiration. Nature 418, 344–348. doi:10.1038/nature00829

Monteiro, P.T., Oliveira, J., Pais, P., Antunes, M., Palma, M., Cavalheiro, M., Galocha, M., Godinho, C.P., Martins, L.C., Bourbon, N., Mota, M.N., Ribeiro, R.A., Viana, R., Sá-Correia, I., Teixeira, M.C., 2020. YEASTRACT+: a portal for cross-species comparative genomics of transcription regulation in yeasts. Nucleic Acids Res. 48, D642–D649. doi:10.1093/nar/gkz859

Ristow, M., Schmeisser, K., 2014. Mitohormesis: Promoting Health and Lifespan by Increased Levels of Reactive Oxygen Species (ROS). Dose Response 12, 288–341. doi:10.2203/dose-response.13-035.Ristow

Wu, Z., Puigserver, P., Andersson, U., Zhang, C., Adelmant, G., Mootha, V., Troy, A., Cinti, S., Lowell, B., Scarpulla, R.C., Spiegelman, B.M., 1999. Mechanisms controlling mitochondrial biogenesis and respiration through the thermogenic coactivator PGC-1. Cell 98, 115–124. doi:10.1016/S0092-8674(00)80611-X

Young, J.C., Hoogenraad, N.J., Hartl, F.U., 2003. Molecular chaperones Hsp90 and Hsp70 deliver preproteins to the mitochondrial import receptor Tom70. Cell 112, 41–50. doi:10.1016/s0092-8674(02)01250-3